# Micropillar arrays, wide window acquisition and AI-based data analysis improve comprehensiveness in multiple proteomic applications

Manuel Matzinger [1,7] ✉, Anna Schmücker [2,3,4], Ramesh Yelagandula [2,5,6], Karel Stejskal[1,2,5], Gabriela Krššáková[1,2,5], Frédéric Berger [2], Karl Mechtler [1,2,5] ✉ & Rupert L. Mayer [1,7] ✉

Comprehensive proteomic analysis is essential to elucidate molecular pathways and protein functions. Despite tremendous progress in proteomics, current studies still suffer from limited proteomic coverage and dynamic range. Here, we utilize micropillar array columns (μPACs) together with wide-window acquisition and the AI-based CHIMERYS search engine to achieve excellent proteomic comprehensiveness for bulk proteomics, affinity purification mass spectrometry and single cell proteomics. Our data show that μPACs identify ≤50% more peptides and ≤24% more proteins, while offering improved throughput, which is critical for large (clinical) proteomics studies. Combining wide precursor isolation widths of m/z 4–12 with the CHIMERYS search engine identified +51–74% and +59–150% more proteins and peptides, respectively, for single cell, co-immunoprecipitation, and multi-species samples over a conventional workflow at well-controlled false discovery rates. The workflow further offers excellent precision, with CVs <7% for low input bulk samples, and accuracy, with deviations <10% from expected fold changes for regular abundance two-proteome mixes. Compared to a conventional workflow, our entire optimized platform discovered 92% more potential interactors in a protein-protein interaction study on the chromatin remodeler Smarca5/Snf2h. These include previously described Smarca5 binding partners and undescribed ones including Arid1a, another chromatin remodeler with key roles in neurodevelopmental and malignant disorders.

Investigating entire proteomes, which define cellular identity is one of the major objectives in the field of proteomics. A comprehensive proteome map is further important to elucidate entire molecular pathways. Improvements in sample preparation techniques and LC-MS/MS instrumentation have enabled the identification of thousands of proteins per sample. Powerful chromatographic separation plays a crucial role in reducing sample complexity in LC-MS/MS analyses and is therefore of key importance to increase proteomic coverage. However, a high degree of separation is typically achieved by using long, time-consuming gradients, that limit throughput as well as sensitivity due to signal dilution and peak broadening[1]. By applying offline fractionation, up to 12,000 proteins have been successfully quantified to

comprehensively map a human cellular system[2]. Fractionation, however is a time-consuming step that might not be feasible for projects subject to limited sample amounts or MS instrument time. These limitations are even more pronounced for single cell proteomic analyses. Although protein identification numbers for single cell shotgun experiments are constantly improved[3,4], these studies usually cover only the most abundant proteins within the cell[4,5]. TMT multiplexing was already used to increase analytical depth for single cell proteomics[6–9] but suffers from a limited dynamic range and missing values across replicates due to the stochasticity of the used data-dependent acquisition (DDA) methods[4]. Label-free approaches employing data-independent acquisition (DIA) methods[10,11] alleviate the data completeness issue but cannot compete with the sample throughput achievable for multiplexed workflows.

This limited analytical depth presents a substantial hurdle that needs to be faced to allow the investigation of critical biological players such as post translational modifications (PTMs), transcription factors or other regulators present at low copy numbers. Missing values across replicates are problematic and particularly common for lower abundant peptide species as their respective precursor ions are only stochastically triggered when using DDA[12]. As dataset sizes increase, missing values become even more prevalent. Modern data analysis strategies attempt to reduce missing values by matching precursor ions at the MS1 level across runs while the remaining empty values are typically imputed by low numbers to allow for statistical testing[13,14]. This approach however introduces additional difficulties in false discovery rate (FDR) estimation and potentially compromises proper quantitative accuracy. Data independent acquisition (DIA), in contrast, acquires all theoretical fragment ion spectra in sequential, predetermined windows[15]. The resulting DIA data is however harder to analyze, as longer cycle times result in fewer datapoints per peak for use in label-free quantification. For multiplexed sample analysis using isobaric labels, simultaneous co-isolation of many precursor ions in DIA mode prevents correct annotation of reporter ions to peptides and hence accurate quantification.

In this study a multitude of approaches is combined to overcome the aforementioned challenges. Advanced chromatographic setups based on microfabricated pillar array columns (µPAC) are benchmarked. In contrast to packed bed columns, these µPAC columns consist of highly ordered pillar arrays that provide exceptionally homogenous flow paths. These homogenous flow paths reduce peak broadening, which yields sharper peaks and thus higher signal intensities[16]. Compared to classical fully porous beads, the superficially porous material on the surface of these micropillars reduces the persistent adsorption of hydrophobic peptides, thereby reducing carryover[17]. One of the benchmarked µPAC columns, a prototype version of the 5.5 cm High-Throughput µPAC Neo HPLC Column, features rectangular-shaped pillars allowing flow path lengths of about 50 cm in very short (5.5 cm) physical column lengths, resulting in low backpressures at nanoflow rates and a broad range of available flow rates up to 2.5 µL/min. These capabilities enable fast column loading and conditioning, which increases sample throughput substantially[17,18]. On the mass spectrometer, the wide-window acquisition (WWA) method merges the strengths of DDA and DIA. WWA uses a broad isolation window (m/z ≥ 4) for data-dependent precursor selection. Similar to DIA, precursor ions close to those selected are co-fragmented, producing chimeric spectra that boost IDs and improve coverage of low-abundance peptides, which would otherwise be missed. WWA is particularly powerful when combined with the AI-driven CHIMERYS search algorithm, which allows confident identification of many peptides from a single chimeric spectrum. Developed by MSAID, the CHIMERYS algorithm makes use of accurate fragment spectrum predictions trained on millions of spectra, which allows to utilize additional spectral properties such as relative signal intensities for drastically improved

identification rates[19]. Combining these innovations within a single analytical platform yields substantially improved sensitivity, throughput, and proteome coverage as well as reduced numbers of missing values. Exemplified for a protein-protein interaction study on the chromatin remodeler Smarca5/Snf2h, more than twice as many proteins can be identified as compared to a standard analytical platform. Accordingly, also 92% more potential interactors are detected including previously described Smarca5-interactors as well as undescribed ones like Arid1a, which was primarily related to chromatin remodeling by Swi/Snf and has been described as a key player in neurodevelopmental and malignant disorders. The proposed LC-MS/MS-based analytical platform could therefore play a central role in many future studies to facilitate the discovery of transformative insights into clinical and biological questions.

## Results
### Combination of µPAC with WWA and AI-driven data analysis results in unprecedented proteomic coverage
In our analytical platform, we combine recent technological innovations in liquid chromatography, mass spectrometry and data analysis to further improve the depth of analysis as illustrated in Fig. 1. We utilize the high versatility offered by the Vanquish Neo LC system, which offers a high flow rate range from a recommended 100 nL/min up to 100 µL/min without the need to install different flow meters. This is particularly helpful for columns that can accommodate high flow rates during sample loading or after analysis during washing and equilibration to reduce overhead times and improve sample throughput directly needed for the analysis of larger biological studies or clinical cohorts.

The use of µPAC columns allows to reduce peak broadening due to minimized Eddy diffusion. These columns can cope with higher flow rates due to their unique design, which reduces backpressures as compared to packed bed columns, making them ideally suited for use on the Vanquish Neo LC system for fast loading, washing and equilibration at higher flow rates.

Mass spectrometric detection is aided by a FAIMS Pro interface, which acts as an ion filter based on charge state, molecular shape, conformation, and size to reduce the influx of undesired, singly charged background ions and thus improves signal to noise ratio of resulting spectra. In contrast to typical DDA approaches, which classically employ narrow precursor isolation windows in the range of 0.7–2 Da, we utilize WWA with precursor isolation windows of 4 Da and wider. We use these wide isolation windows as the CHIMERYS downstream data analysis tool can accurately and sensitively identify a high number of peptides from chimeric spectra. CHIMERYS typically surpasses the number of protein and peptide IDs detected by classical search engines substantially due to the inclusion of relative fragment intensities and other parameters often neglected in other search engines. CHIMERYS is embedded in Proteome Discoverer 3.0, which provides raw data pre-processing as well as further data analysis capabilities and statistical options.

This advanced analysis platform was characterized by a multitude of different experiments, which are listed in Table 1. To this end, a number of different samples were measured including 12.5 ng K562 digests with added standard peptides (Q4L) showcasing the excellent performance of µPAC columns over packed bed columns in our hands (Fig. 2). For comparison of different µPAC columns to each other, a triple proteome mix of tryptic HeLa, yeast, and *E. coli* digests was analyzed to highlight the different strengths of the three µPAC columns at various gradient lengths and input amounts (Fig. 3). These analyses also showed the tremendous throughput that can be achieved with the 5.5 cm High-Throughput µPAC Neo HPLC Column (Fig. 4 and Table 2). 200 ng triple proteome mix was also analyzed in parallel with DDA and WWA (isolation width 4) and data analysis carried out with both MS Amanda 2.0 and CHIMERYS, which demonstrated substantially higher numbers of peptide and protein IDs when applying WWA and CHIMERYS in combination (Fig. 5). Furthermore,

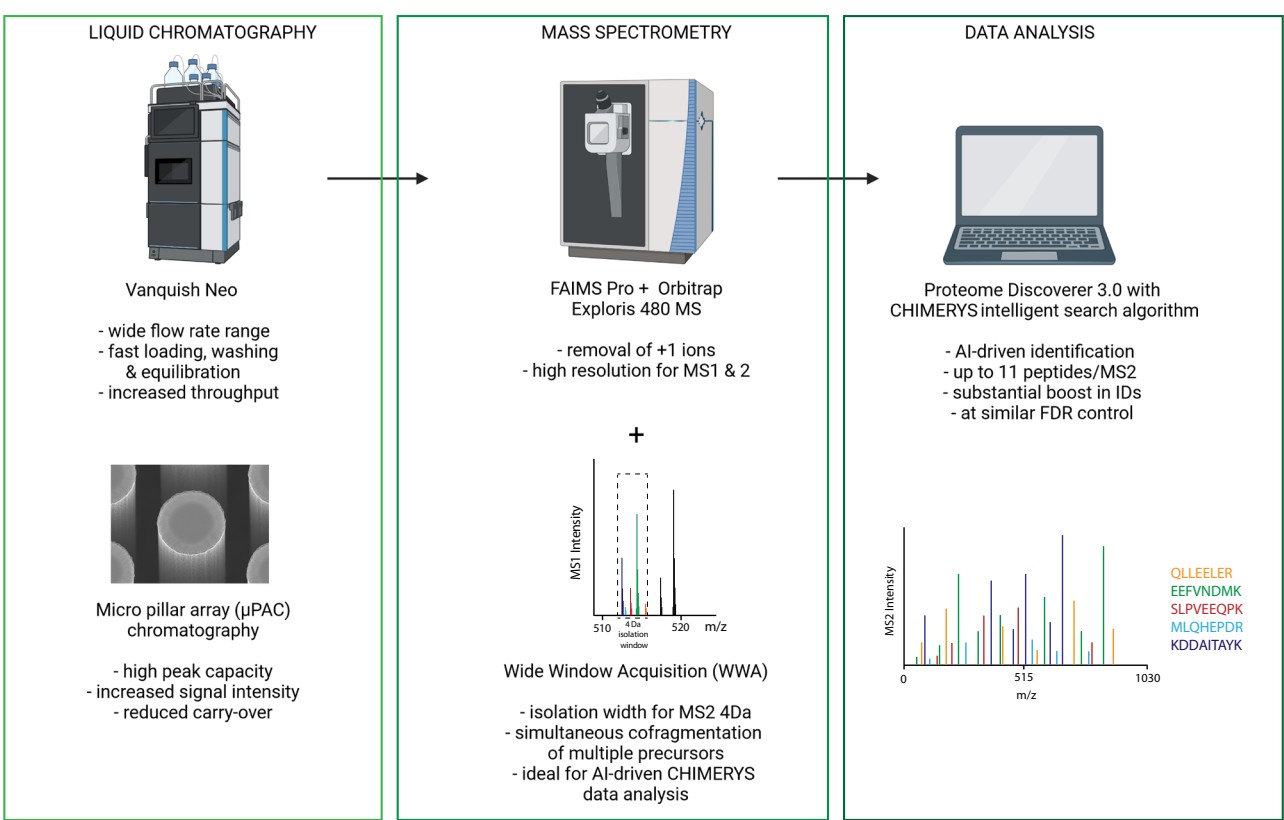

**Fig. 1 | Combination of cutting-edge technological advancements in liquid chromatography, mass spectrometry and data analysis to achieve unprecedented proteomic depth.** Scheme of the employed workflow including the use of the 2021-launched Vanquish Neo LC system and innovative μPAC columns, next to state-of-the-art FAIMS Pro and Orbitrap Exploris 480 mass spectrometry in conjunction with WWA as well as the versatile Proteome Discoverer 3.0 platform and the AI-driven CHIMERYS search algorithm. Created with BioRender.com.

**Table 1 | Experimental overview to assess the advanced platform performance in part and overall**

| Platform Constituent Assessed | Question | Display item |
|---|---|---|
| LC setup | Packed columns vs μPAC | Fig. 2 |
| | Comparison of different μPACs | Fig. 3 |
| | Protein IDs/min and throughput | Fig. 4 and Table 2 |
| MS acquisition | WWA vs DDA | Fig. 5A, B |
| | Isolation width evaluation for WWA | Fig. 5D, E |
| Search engine | CHIMERYS vs classical search engine | Fig. 5A–C |
| | Entrapment experiment for FDR estimation | Fig. 5C |
| Application – low input & single cells | Influence of WWA + /– MBR on protein IDs and quantification for (very) low inputs | Fig. 6A–D |
| Application - two-species proteome mix | Evaluation of quantitative accuracy for DDA and WWA using CHIMERYS | Supplementary Fig. 6 |
| Application – AP-MS – entire workflow | Impact of entire workflow (μPAC, WWA, CHIMERYS) on identification of additional interaction partners in AP-MS | Fig. 7 and Table 3 |

WWA and CHIMERYS recapitulated nearly all proteins that were identified with the classical approach as well as 50% additional proteins. HeLa cell lysates at various input amounts were measured using different precursor isolation widths to assess the ideal isolation width revealing that lower input amounts require wider isolation windows for optimal results. False discovery rate control of CHIMERYS for DDA and WWA was assessed by entrapment experiments using mouse affinity purification samples showcasing excellent protein FDR control at 1% and above. To assess the suitability of the pipeline for very low input amounts, samples of single and 40 HeLa cells were measured as well as 250 pg and 10 ng of HeLa bulk. This demonstrated contemporary sensitivity and high quantitative reproducibility of the advanced pipeline for low input amounts. (Fig. 6A–D)

In addition, the quantitative accuracy of the pipeline was assessed by measuring double proteome mixes containing different ratios of HeLa and yeast digests. These results as demonstrated in Supplementary Fig. 6 show excellent accuracy for WWA with <10% deviation from the expected fold changes. Previously published AP-MS data by Furlan et al. was reanalyzed using CHIMERYS and resulted in more proteins and peptides with a similar or slightly larger number of identified interactors (Fig. 7). Furthermore, mouse embryonic stem cells were probed for interactors of the chromatin remodeler Smarca5, and the resulting samples analyzed with a classical and the advanced proteomics pipeline. Substantially more interactors were identified for the advanced pipeline, of which many had been previously described. (Fig. 7 and Table 3).

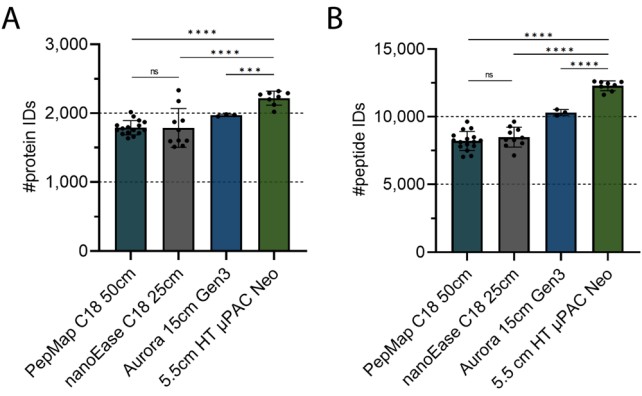

**Fig. 2 | Benchmarking packed vs µPAC columns.** 12.5 ng of a K562 QC mix were injected each using a trap-and-elute setting. Peptides were separated over 30 min using a linear gradient over 30 min from 1–35% buffer B (80% acetonitrile, 0.1% formic acid). Data acquisition using a standard DDA method with an isolation window of m/z 1 and data analysis using CHIMERYS at 1% FDR on peptide and protein level, $n = 16$, 10, 3 and 8 technical replicates for the PepMap, nanoEase, Aurora and µPAC runs, respectively. Bars indicate means, while error bars indicate standard deviations. **A** Protein and (**B**) peptide identifications are visualized. Statistical significance between means of different groups was assessed by two-tailed, unpaired Student $t$ tests for all comparisons but those involving nanoEase and Aurora protein IDs due to differences in variance, for which two-tailed, unpaired Welch's tests were performed. ns not significant, ***$p \leq 0.001$, ****$p \leq 0.0001$. Exact $p$ values for (**A**) µPAC Neo vs PepMap <0.0001, µPAC Neo vs nanoEase 0.0008, nanoEase vs PepMap 0.9715, µPAC Neo vs Aurora 0.0002, and (**B**) µPAC Neo vs PepMap <0.0001, µPAC Neo vs nanoEase <0.0001, nanoEase vs PepMap 0.3443, µPAC Neo vs Aurora <0.0001. Source data are provided as a Source Data file.

## Micro pillar array columns outperform packed bed columns

To assess column performance, we used a QC standard mix from a K562 cell digest including reference peptides in use within the Core for Life alliance[20] to benchmark a prototype version of the 5.5 cm High-Throughput µPAC Neo HPLC Column against classical packed bed columns from Thermo (PepMap, 50 cm bed length), Waters (NanoEase, 25 cm bed length) and IonOpticks (Aurora Elite Column 15 cm bed length). As illustrated in Fig. 2, the 5.5 cm brick prototype column outperforms its competitors by increasing protein and peptide IDs by ≤24% and ≤50%, respectively. Based on these results, we decided to further investigate and benchmark different µPAC columns, gradient lengths, and input amounts to obtain a comprehensive overview of column performance for a broad variety of conditions.

## Different µPAC columns optimally facilitate a variety of proteomic applications and allow the identification of more than 10,000 proteins from a single run

To compare the performance and best suited application per column, three different µPAC columns were assessed including a prototype version of the 5.5 cm High-Throughput µPAC Neo HPLC Column, a 50 cm µPAC Neo HPLC Column as well as a prototype version of the 110 cm µPAC Neo HPLC Column. Triple proteome mixes of tryptic HeLa, yeast, and *E. coli* digests were prepared at a ratio of 8:1:1 in 0.1% TFA at three different peptide concentrations 10 ng/µL, 100 ng/µL and 400 ng/µL. 0.5–1 µL of the lysate mix was injected directly onto the analytical column without a trapping column using the Vanquish Neo LC system. Columns were grounded via the metal housing of the LC system to prevent charging of the pillar array to avoid losses in separation power. Injection amounts were varied between 10 ng and 400 ng of total peptide material and gradient lengths were assessed up to 120 min. While the 5.5 cm Neo column and the 110 cm Neo column were connected to the Nanospray Flex PepSep sprayer via a custom-made connecting capillary (20 µm ID x 360 µm OD, length 20 cm), the

50 cm Neo column was connected directly to the sprayer via its nanoViper fittings.

Depending on the column length and the available flow rates, different gradient lengths per column type were tested including 5–60 min for the 5.5 cm Neo column, and 30–120 min for the 50 cm Neo and the 110 cm Neo column. The column volumes for the 5.5 cm and 50 cm column are identical at 1.5 µL, while the 110 cm column has 4.5 µL column volume. Figure 3A visualizes the protein identifications obtained on average from these runs with a minimum of 1934 protein IDs for the 5 min gradient on the 5.5 cm Neo column and a maximum of 10,487 protein IDs for the 50 cm Neo column. For all measurements excellent reproducibility was achieved as indicated by the small error bars.

Unsurprisingly, the 120 min gradient with the two long columns achieved the greatest proteome depth with over 10,000 protein IDs on average for 400 ng injection amount. As expected, a general trend with higher injection amounts and longer gradient lengths towards higher proteome coverage was observed for all columns. The striking protein ID gap between an injection amount of 10 ng and 50 ng had not been reported by earlier studies in our lab, but might largely be attributed to the rather short MS2 maximum injection times of 23 ms used here as demonstrated in Supplementary Fig. 1[17]. In addition, increasing the peptide load led to a less pronounced increase in protein IDs. Only for longer gradients, the protein ID gap between different peptide loads showed a wider spread. When comparing the columns for the 30 min and 60 min gradients, it became evident that the 50 cm Neo column performed best with the highest number of protein IDs. Particularly at 10 ng with the 30 min gradient it outperformed the 5.5 cm and 110 cm column with 4222 over 3506 and 3316 protein IDs, respectively. At higher sample loads and longer gradients this gap was reduced, as shown for the 60 min gradient at 400 ng injection amount at 9309 protein IDs for the 50 cm Neo column in contrast to 9037 and 9058 protein IDs for the 5.5 and 110 cm column, respectively. Eventually, for the 120 min gradient at 400 ng, there was no difference in protein IDs between the 50 cm Neo column and the 110 cm column (10,421 protein IDs on average for both).

This increased sensitivity at low sample amounts but similar performance at higher sample loads for the 50 cm Neo column could be explained by the architecture of the column and the nanoviper fittings of the 50 cm Neo column (that was not equal the other two columns) allowing direct coupling to the valve and the pepsep sprayer with reduced post-column volume reducing peak broadening. As expected, this was particularly pivotal at low concentrations and became less important at high sample loads. The 5.5 cm and 110 cm columns perform similarly for the 30 and 60 min gradients. While the longer 110 cm column intuitively seems to be better suited for higher sample loads and longer gradients, the 5.5 cm column can also be readily used for very short runs utilizing gradient times of only 10 or even 5 min. We, therefore, consider the 50 cm Neo column ideal as an all-round column that is ideally suited for lower as well as intermediate sample loads of 10–500 ng and shorter to intermediate gradient lengths around 15–120 min. The 5.5 cm column in contrast seems to be best applied for shorter gradients due to the short physical length of the column and the high maximum flow rate up to 2.5 µL/min. The 110 cm column did not reach its sweet spot in our tests so far but shows a tendency towards longer gradients of 120 min or longer and high sample loads of ≥400 ng. Also, the column volume of the 110 cm column is substantially larger at 4.5 µL than the 1.5 µL of the 5.5 cm and 50 cm column mandating longer washing and equilibration times leading to increased overhead times.

## Unique column architecture of 5.5 cm column enables reduced run-to-run times resulting in superior sample throughput

While typically longer packed bed capillary columns have very limited maximum flow rates of up to ~500 nL/min, the 5.5 cm column offers a

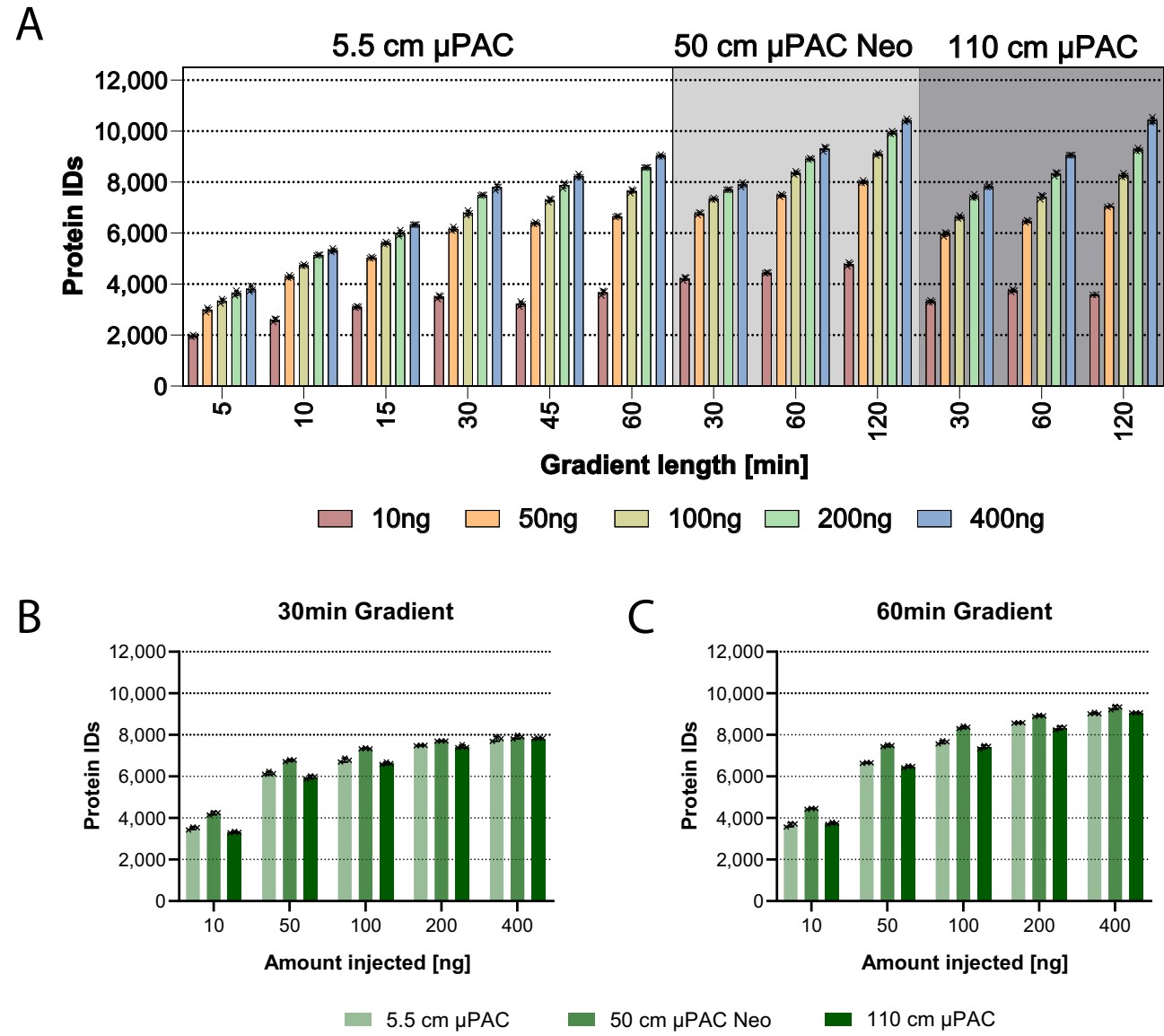

**Fig. 3 | Benchmark of µPAC column designs and gradient lengths. A** Bars indicate average number of identified proteins at 1% FDR on protein level when acquiring using different column lengths, gradient times and input amounts indicated of a 8:1:1 H:Y:E proteome mix. Error bars indicate standard deviations, $n = 2$ technical replicates for 120 min 10 ng and 50 ng $n = 3$ technical replicates for all other conditions. Bars indicate means per condition. **B** Comparison of all columns for 30 min gradient from 10–400 ng peptide load. **C** Comparison of all columns for 60 min gradient from 10–400 ng peptide load. Data for (**B**) and (**C**) is already presented in panel A but displayed differently for easier visual comparison. Source data are provided as a Source Data file.

maximum flow of 2.5 µL/min. This column features rectangular-shaped micropillars in the form of bricks resulting in highly orthogonally prolonged flow paths and relatively large interpillar distances. This allows for highly efficient sample loading, washing, and conditioning in direct injection mode without the use of a trapping column. This substantially reduces the analysis overhead times and leads to more efficient use of the mass spectrometer as indicated in Fig. 4A. The analysis of protein IDs/min run-to-run time for a 30 min gradient method revealed that, while the 50 cm Neo and the 110 cm column could achieve a best of 108 and 86 protein IDs/min run-to-run time, respectively, the 5.5 cm column delivered substantially more IDs/min at 147 representing a gain of 36% and 71% over the 50 and 110 cm column, respectively. When further cutting the gradient length down to 5 min, using the shortest column (Fig. 4B) acquisition of up to 250 protein IDs/min and up to 96 sample injections per day are achievable at a maximum analytical depth of 4000 protein IDs for the triple proteome samples at 400 ng input. When reducing the throughput to

69 and 44 samples/day, up to 5400 and 6300 proteins could be identified, respectively with the 5.5 cm column for 400 ng. At a throughput of 14 samples/day, a best of 9000 proteins could be identified.

For equilibration of all columns, two column volumes with 1% B were selected amounting to 3 µL for the 5.5 cm and 50 cm columns and 9 µL for the 110 cm column. Further details on the gradients can be retrieved from Supplementary Data 1.

Particularly for short or very short gradients of ≤30 min, reduced overhead times become more and more impactful as the overhead times remain stable while the total analysis time is reduced thereby consuming a substantial relative fraction of the total analysis time as illustrated in Table 2. At a flow rate during the analysis of 300 nL/min, the 110 cm column offers little room for speeding up sample loading and equilibration as the maximum flow rate for this column was specified during the time of testing around 400 nL/min. Equilibration for all columns and gradient lengths was performed so that two column

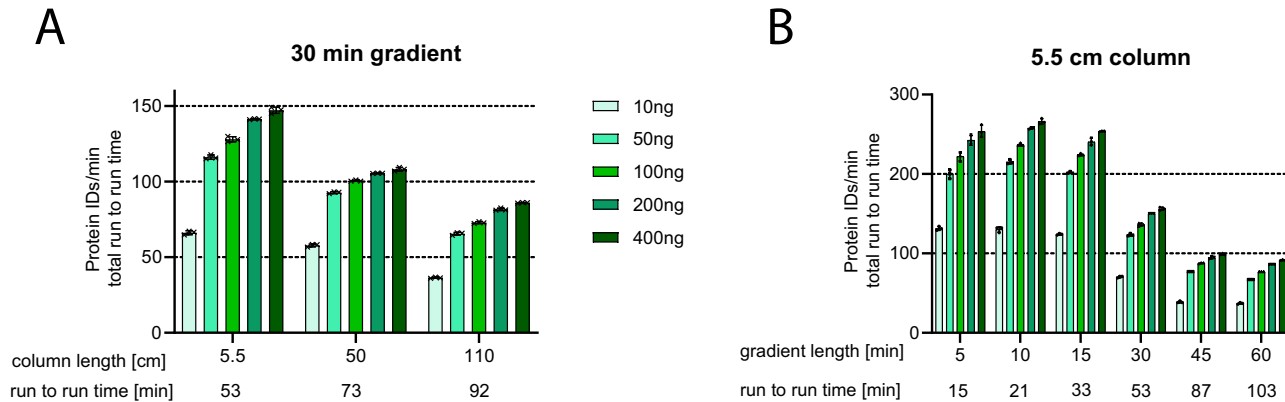

**Fig. 4 | Assessment of protein IDs/min.** Identified proteins at 1% peptide and protein FDR were normalized to the complete time between each injection ("run-to-run time") to evaluate how efficiently the mass spectrometer was utilized over the entire duration of the run. *n* = 3 technical replicates for each condition, error bars indicate standard deviations, bars indicate means per condition. **A** Comparison of column types using a fixed gradient length. **B** Comparison of different gradient lengths using the 5.5 cm column. Source data are provided as a Source Data file.

**Table 2 | Different column lengths and geometries result in substantially variable overhead times**

| Gradient length [min] | Wash time [min] | Total over-head [min] | Run-to-run time [min] | Samplesper day no washes or QCs | % over-head of run-to-runtime | Total over-head [min] | Run-to-run time [min] | Samples per day no washes or QCs | % over-head of run-to-runtime |
|---|---|---|---|---|---|---|---|---|---|
| all columns | all columns | 5.5 cm column | 5.5 cm column | 5.5 cm column | 5.5 cm column | 110 cm column | 110 cm column | 110 cm column | 110 cm column |
| 5 | 3.5 | 6.5 | 15 | 96 | 43 | 47 | - | - | - |
| 10 | 4.0 | 7.0 | 21 | 69 | 33 | 47 | - | - | - |
| 15 | 10 | 8.0 | 33 | 44 | 24 | 47 | - | - | - |
| 30 | 15 | 8.0 | 53 | 27 | 15 | 47 | 92 | 16 | 51 |
| 60 | 15 | 28* | 103* | 14* | 27* | 47 | 122 | 12 | 39 |
| 120 | 15 | - | - | - | - | 47 | 182 | 8 | 26 |

All gradients used for the 5.5 cm column (except for the 60 min gradient) are presented with optimized overhead times featuring 2.5 μL/min maximum flow rate for loading and equilibration with pressure control on at 300 bar maximum. In comparison, the 110 cm column could only be operated at ≤400 nL/min due to higher backpressure and consequently reduced flow rate range. Overhead times include sample pickup and loading as well as equilibration after washing. Due to the high maximum flow rate for the 5.5 cm column, overhead times can be dramatically faster allowing sample throughputs of up to 96 samples per day. The 110 cm column further suffers from a three times higher column volume exacerbating the overhead times due to longer equilibration. Both columns were equilibrated after washing with 2 column volumes of 1% B. Of note, 5, 10 and 15 min gradients were not realized using the 110 cm column due to its limited flow rate and the 120 min gradient was not realized on the 5.5 cm column.
*Loading was performed at a maximum flow rate of 400 nL/min.

volumes of 1% B were passed over the column before the start of the next injection. The recently commercially available version of the 110 cm μPAC column allows a maximum flow rate of 750 nL/min facilitating faster loading and equilibration. The 5, 10 and 15 min gradient lengths were not measured on the 110 cm column due to the low maximum flow rate. In contrast, the 5.5 cm prototype column allows flow rates up to 2500 nL/min, offering considerably accelerated sample loading and column equilibration yielding up to 96 sample runs per day for very short 5 min gradients.

**The AI-driven search engine CHIMERYS substantially improves analysis depth at well controlled FDR using sample input dependent ideal isolation widths**

For better utilization of all information provided within fragment spectra, we evaluated the use of the AI-driven, groundbreaking search engine CHIMERYS. Next to the m/z values of peptide fragments, CHIMERYS can harness additional spectral information such as relative abundance of fragment signal, which allows the accurate identification of numerous peptides from highly chimeric fragmentation spectra resulting from our proteome-mix representing a highly complex sample. Figure 5A clearly highlights that CHIMERYS offers improved proteomic depth compared to a more classical state-of-the-art search engine with an additional 68% peptide IDs and 32% protein IDs for a typical m/z 1 isolation window. Even more pronounced was the improvement on the identification rate assigning 2.6 times as many

PSMs to MS/MS spectra in comparison to MS Amanda 2.0[21,22]. Using a wider precursor isolation window of 4 m/z, which is further referred to as WWA, richer and more complex MS2 spectra could be generated, further favoring the advantage of CHIMERYS to decipher highly chimeric fragmentation spectra. This boosted the identification rate of CHIMERYS to around 125% meaning that on average there was more than one PSM per MS/MS spectrum identified, corresponding to a 4.6-fold boost over the classical search engine employed, while peptide IDs were boosted by 113% and protein IDs by 41%. Figure 5B indicates that these additional protein IDs substantially overlap with the results from MS Amanda 2.0, thereby conserving confident IDs, while detecting low abundance proteins. Of note, we further observed that CHIMERYS in combination with WWA improved dynamic range of tryptic peptides. To this end, PSMs from triple proteome samples containing human, yeast and *E. coli* were ranked according to intensity and assigned to a PSM index from the lowest abundant PSM starting at PSM index 1 up to the highest abundant PSM with the highest PSM index as displayed in Supplementary Fig. 2. This plot visualizes that the combination of CHIMERYS and WWA unlocks a broader dynamic range of peptides for identification, which in turn allows the identification of less abundant proteins.

For the assessment of the FDR control of CHIMERYS, entrapment experiments were conducted searching raw data with a target database and an additional decoy database on top, which allowed us to quantify the number of actual false positive

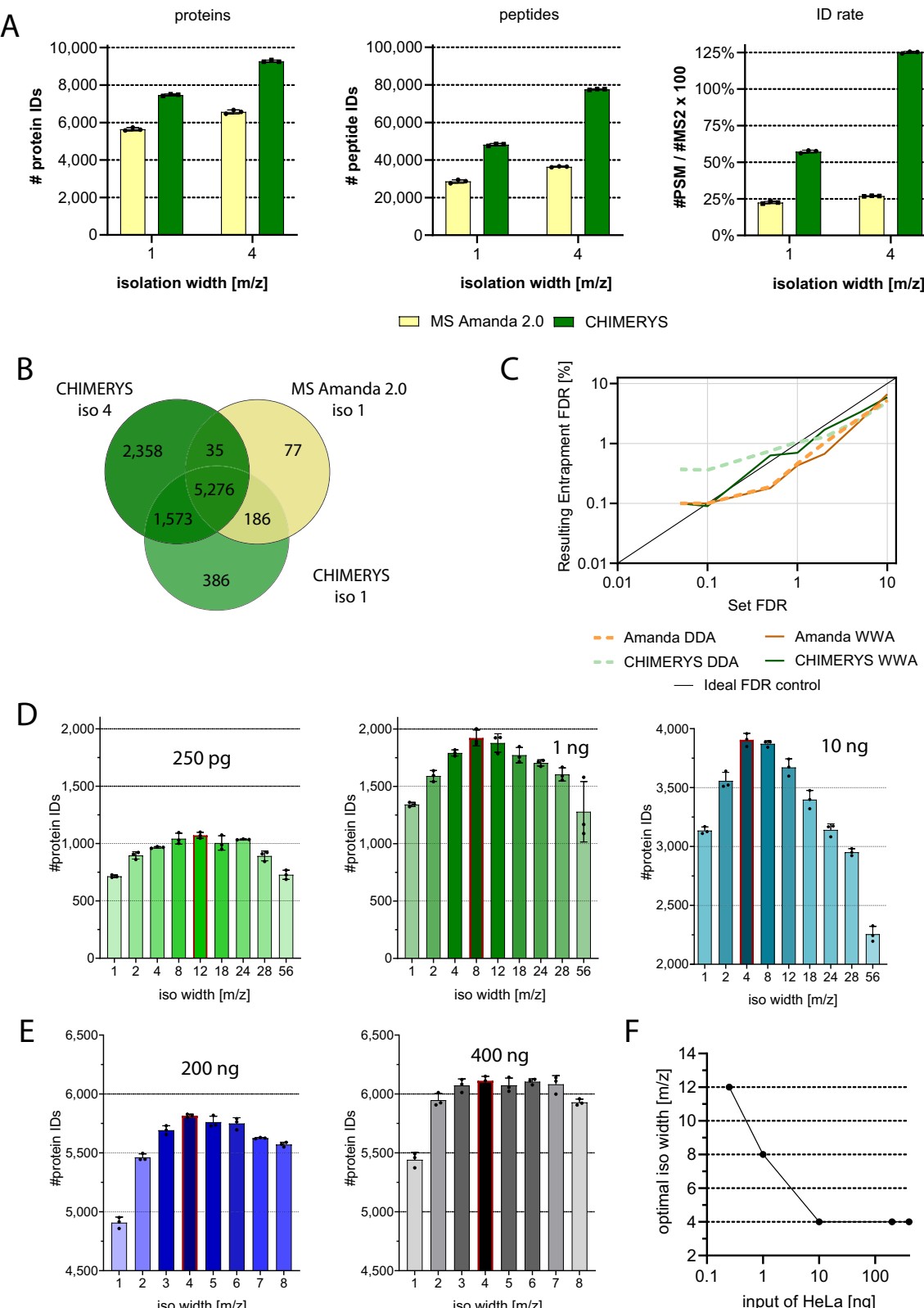

hits for several chosen FDR settings. To this end, two mouse co-immunoprecipitation runs, one DDA and one WWA run, were searched against the mouse Uniprot reference database (2022-03-04; 21,962 sequences and 11,728,099 residues) as well as a decoy database created from the target database as described by ref. 23. In short, for the decoy database, tryptic in silico digestion of the target database was performed without any

missed cleavages allowed. The resulting peptide sequences were grouped into short peptides (<6 amino acids) and unique peptides. Amino acids within any unique peptide were then shuffled with the C-terminal amino acid being conserved. Shuffled peptides with sequences identical to unique peptide sequences of the target database were removed. The remaining shuffled peptides and the short peptides, which were not shuffled, were reassigned

**Fig. 5 | AI-driven search engine CHIMERYS boosts protein IDs substantially at well controlled FDR using wide window acquisition. A** 200 ng proteome-mix H:Y:E = 8:1:1 were separated over 120 min using the 110 cm column prior to MS acquisition and data analysis with the indicated strategy. Typical DDA measurements show improved ID rates on peptide and protein level using CHIMERYS. This boost is even more pronounced when using WWA with precursor isolation widths of m/z 4. *n* = 3 technical replicates, error bars represent standard deviation, bars indicate means. **B** CHIMERYS identifies the same proteins as MS Amanda 2.0, and additional proteins improving proteomic coverage of the sample under analysis. Venn diagram showing average protein ID numbers and overlap. *n* = 1 per condition. **C** Two mouse co-immunoprecipitation samples were searched with a target database and an additional custom-made decoy database to estimate the FDR control of CHIMERYS and MS Amanda 2.0. The results demonstrate excellent FDR control for both software tools and both the DDA as well as the WWA samples from 1% upwards. Peptide and PSM FDRs are illustrated in Supplementary Fig. 3. *n* = 1 per condition. **D**, **E** Different isolation window sizes were tested to identify the most well-suited precursor isolation window size for maximum identifications using tryptic HeLa digests. Red bars indicate isolation width with most IDs. Color shading reflects IDs: lightest colors indicate lowest and darkest colors highest ID counts/input. *n* = 3 technical replicates, error bars represent standard deviation, bars indicate means. **D** low sample input such as 250 pg up to 10 ng measured on the 5.5 cm column and (**E**) standard sample input from 200 to 400 ng measured on the 50 cm µPAC column. **F** Optimal isolation width is plotted against injected sample input. *n* = 1 per condition. Source data are provided as a Source Data file.

to their respective position within the protein of origin to form the final shuffled decoy database.

As depicted in Fig. 5C, the FDR control starting from 1% is demonstrated to be excellent for both software tools and both acquisition modes on the protein level as well as the peptide and PSM level (see Supplementary Fig. 3. For CHIMERYS, the FDR control below 1% seems to be slightly better for the WWA run than for the DDA run. When comparing MS Amanda 2.0 with CHIMERYS, a slightly better, more conservative, FDR control is observed below 1%. Above an FDR of 1%, however, results for both search algorithms start to coalesce and become virtually identical at an FDR of 5%. For MS Amanda 2.0 nearly no differences can be observed for WWA vs DDA.

We tested an array of different precursor isolation widths as illustrated in Fig. 5D–F for different input amounts of tryptic HeLa digest from 250 pg up to 400 ng. Deducing from these data, we could assign m/z 4 as ideal isolation width for maximum protein IDs for standard injection amounts (200–400 ng), while the best isolation width for low input samples was around m/z 8–12. Of note, this is in line with observations made by ref. 11 using a DIA approach. They report that the optimal isolation window size is wider with lowered input amount. Considering the reduced complexity that is to be expected from low input samples, it seems intuitive that wider isolation windows are beneficial to reach similar levels of complexity in the fragmentation spectra. Indeed, our data as well as the data of others[24] support the advantage of broader isolation windows for lowered input amounts (see Fig. 7C). This trend is even more pronounced on the peptide level or when investigating the number of identified peptides per spectrum (identification rate) (Supplementary Fig. 4). Hence, for best utilization of the CHIMERYS search engine it appears beneficial to optimize the precursor isolation window according to sample complexity and injection amount to obtain optimal results.

## WWA delivers accurate and precise protein quantifications and increases protein identifications and quantifications for true single cell measurements as compared to DDA

Single HeLa cells as well as 40 cells were isolated into individual wells of a 384 well plate, digested with trypsin applying high humidity and rehydration at 50 °C and digests diluted to 3.5 µL with 0.1% TFA and 5% DMSO as previously described by our group[25]. Cell lysates were analyzed either using DDA with an isolation window of m/z 1 or with WWA applying an isolation window of m/z 12 according to the optimal WWA isolation width for 250 pg HeLa determined earlier. 250 pg and 10 ng HeLa bulk samples were injected also using identical analysis setting. Raw files were first searched individually without match-between-runs (MBR) to determine the respective numbers of protein identifications (Fig. 6A), which yielded substantially enhanced average numbers of protein identifications for WWA over DDA for 250 pg and 10 ng HeLa bulk samples (+74.4% and +20.2%) as well as single cell and 40 cell samples (+78.3% and +42.7%). On average, 536 and 955 proteins were identified for single cells using DDA and WWA, respectively, without

MBR. Particularly the number of proteins identified for single WWA measurements without matching is on par with other current label-free single cell proteomics workflows[25–27]. For Fig. 6B, raw files were further grouped according to acquisition strategy and sample type (DDA 250 pg + DDA 10 ng, WWA 250 pg + WWA 10 ng, DDA single cells + DDA 40 cells and WWA single cells + WWA 40 cells) and searched in batch mode enabling MBR and the number of quantified proteins assessed. Similar to the number of identified proteins, the number of quantified proteins was increased for single cell and 250 pg bulk samples when using WWA. With matching, 923 and 996 proteins were successfully quantified from DDA and WWA for single cell samples, respectively. The application of MBR, however, reduced the benefits of WWA for 250 pg and single cell inputs resulting in minor improvements only. The larger inputs of 40 cells and 10 ng were affected less and WWA resulted in markedly more quantified proteins with and without matching. The reproducibility of quantification was assessed via calculation of coefficients of variation (CVs) for all proteins successfully quantified in the three samples per condition with the most quantified proteins. Without MBR, (Fig. 6C, Supplementary Data 2), WWA resulted in very similar CVs only affording slightly lower CVs for 250 pg inputs. With MBR, (Fig. 6D, Supplementary Data 2) in contrast, WWA resulted in significantly lower CVs for all different input amounts. We assessed whether this could be caused by a reduced cycle time and a correspondingly higher number of datapoints over the peak, but no difference was identified (Supplementary Fig. 5). We therefore hypothesize that the additional number of peptides per protein afforded by WWA stabilized protein quantities, which seems to take effect only when normalization between runs and matching is applied.

Quantitative accuracy of WWA in combination with CHIMERYS was assessed by double proteome mixes of HeLa and yeast digests as visualized in Supplementary Fig. 6 (see also Supplementary Data 3). The total peptide amount per injection was kept constant at 200 ng, with varying contributions per species including 200 ng, 150 ng, 100 ng, 50 ng and 0 ng resulting in 5 different samples that were measured in triplicates using an isolation width of m/z 1 and 4. All DDA and WWA runs were searched in two separate searches using MBR. Average intensities per protein were calculated and compared within the same species between different input amounts. Comparison of the medians of measured fold changes against expected fold changes resulted in small deviations from the expected fold changes in the range of 0.2–9.8%. WWA resulted in slightly improved quantitative accuracy for all comparisons, which was more evident and statistically significant for yeast proteins and less pronounced for human proteins. Both datasets, single cell and low input HeLa samples, as well as the double proteome mix suggest that WWA together with the CHIMERYS search engine deliver accurate and reproducible quantitative results.

## Investigation of AP-MS samples with the optimized analysis platform recovers known interactors and identifies undescribed potential interactors

To assess whether CHIMERYS and apQuant are able to recapitulate well known protein-protein interactions from low input samples, AP-

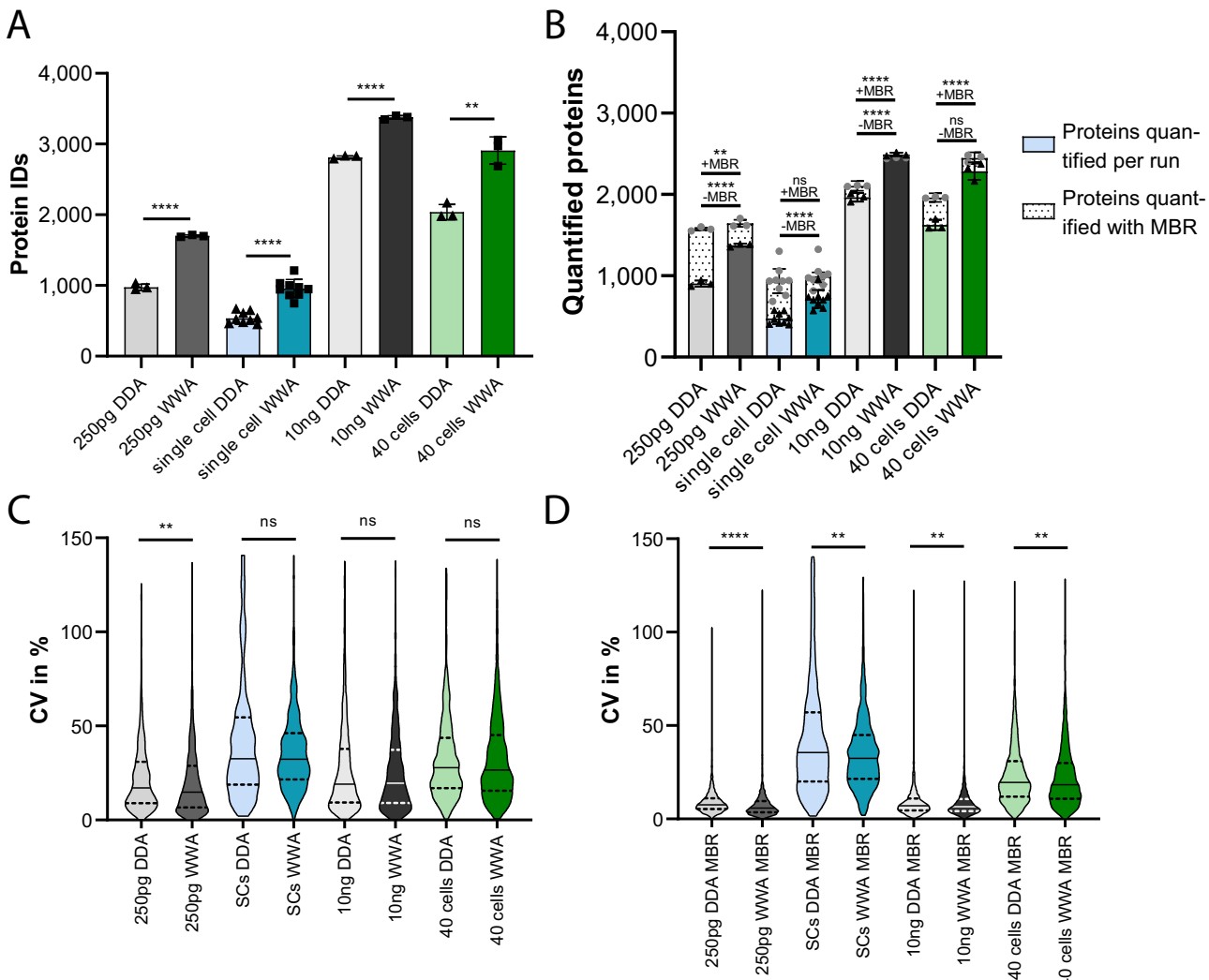

**Fig. 6 | WWA boosts single cell protein IDs and offers precise protein quantification.** HeLa single and 40 cell digests as well as 250 pg and 10 ng bulk digests were recorded using DDA/WWA and analyzed via CHIMERYS and apQuant either file-by-file or in batches using match-between-runs (MBR). Batches were grouped by acquisition and sample type yielding four batches: (i) 250 pg & 10 ng HeLa bulk DDA, (ii) 250 pg & 10 ng HeLa bulk WWA, (iii) HeLa single & 40 cells DDA, and iv) HeLa single & 40 cells WWA. **A** shows protein IDs/run without MBR, while (**B**) depicts quantified proteins without (solid bars) or with MBR (dotted bars). Black triangles indicate quantified proteins/run and gray circles quantified proteins with MBR, $n = 9$ for single cells, $n = 3$ for 40 cells, 250 pg and 10 ng bulk digests. **A, B** Unpaired, two-sided Student $t$ testing was performed except for 40 cells without MBR for which Mann–Whitney testing was employed (non-normality of data). Bars represent means, while error bars indicate standard deviations. **A** $p$ value for 40 cells DDA vs WWA 0.0025, other $p$ values < 0.0001. **B** $p$ values without MBR left to right <0.0001, <0.0001, <0.0001, 0.1000, $p$ values with MBR left to right 0.0064, 0.3537, <0.0001, <0.0001. **C, D** Coefficients of variation (CVs) of proteins quantified in the three replicates with the most quantified proteins/condition were assessed and their distribution plotted. While file-by-file analyses (**C**) resulted in similar CVs for both DDA and WWA, MBR (**D**) yielded slightly reduced CVs for all sample types. Full lines represent median CVs, dashed lines indicate quartiles. No data set passed Shapiro–Wilk normality mandating two-sided Mann–Whitney testing. Protein CV numbers/condition from left to right in graph (**C**) 694, 1055, 312, 565, 1,423, 1,896, 1,114 and 1,641 and in graph (**D**) 1504, 1579, 744, 826, 2068, 2384, 1854 and 2266. $p$ values from left to right (**C**) 0.001, 0.5242, 0.6389, 0.4294 (**D**) <0.0001, 0.0013, 0.0046, 0.0025 (**A–D**) *$p \leq 0.05$, **$p \leq 0.01$, ***$p \leq 0.001$, ****$p \leq 0.0001$. ns $p > 0.05$. Source data are provided as a Source Data file.

MS data previously published by Furlan et al. in 2019 was reanalyzed with CHIMERYS and apQuant[28]. Furlan et al. describe a chip-based, highly sensitive approach for the identification of protein-protein interactions from low input samples consisting of as little as 12,000 cells utilizing MaxQuant for data analysis. Data for two different baits (SMC1A and CDK8) and three different cell inputs (4 µg and 12,000 cells for SMC1A, 25,000 cells for CDK8) were reanalyzed with the same bioinformatic workflow as used for the Smarca5 experiment. Only for 4 µg SMC1A apQuant normalization was turned off as high background of trypsin in the control samples otherwise led to a strong distortion of enriched proteins. Applying the CHIMERYS-based bioinformatic workflow, 76–232% more proteins per experiment (Fig. 7A) were

quantified. As indicated in Fig. 7B and Supplementary Data 4, also the average peptide coverage/protein could be improved for all identified proteins as well as for interactors by 7–18% and 11–37%, respectively. Interestingly, the highest gains were observed for the two low input experiments with 12,000 and 25,000 cells. As demonstrated in Fig. 7C and Supplementary Data 4, all interactors of SMC1A reported by Furlan et al. at 1% FDR could be reproduced, while ~75% of the CDK8 interactors could be recapitulated with the CHIMERYS-based workflow. At 1% FDR, MaxQuant and CHIMERYS uniquely identified 5 and 7 interactors, respectively, most of which were well described CDK8 interactors. These results confirm the high validity of the CHIMERYS-based data analysis workflow, particularly when considering the low-

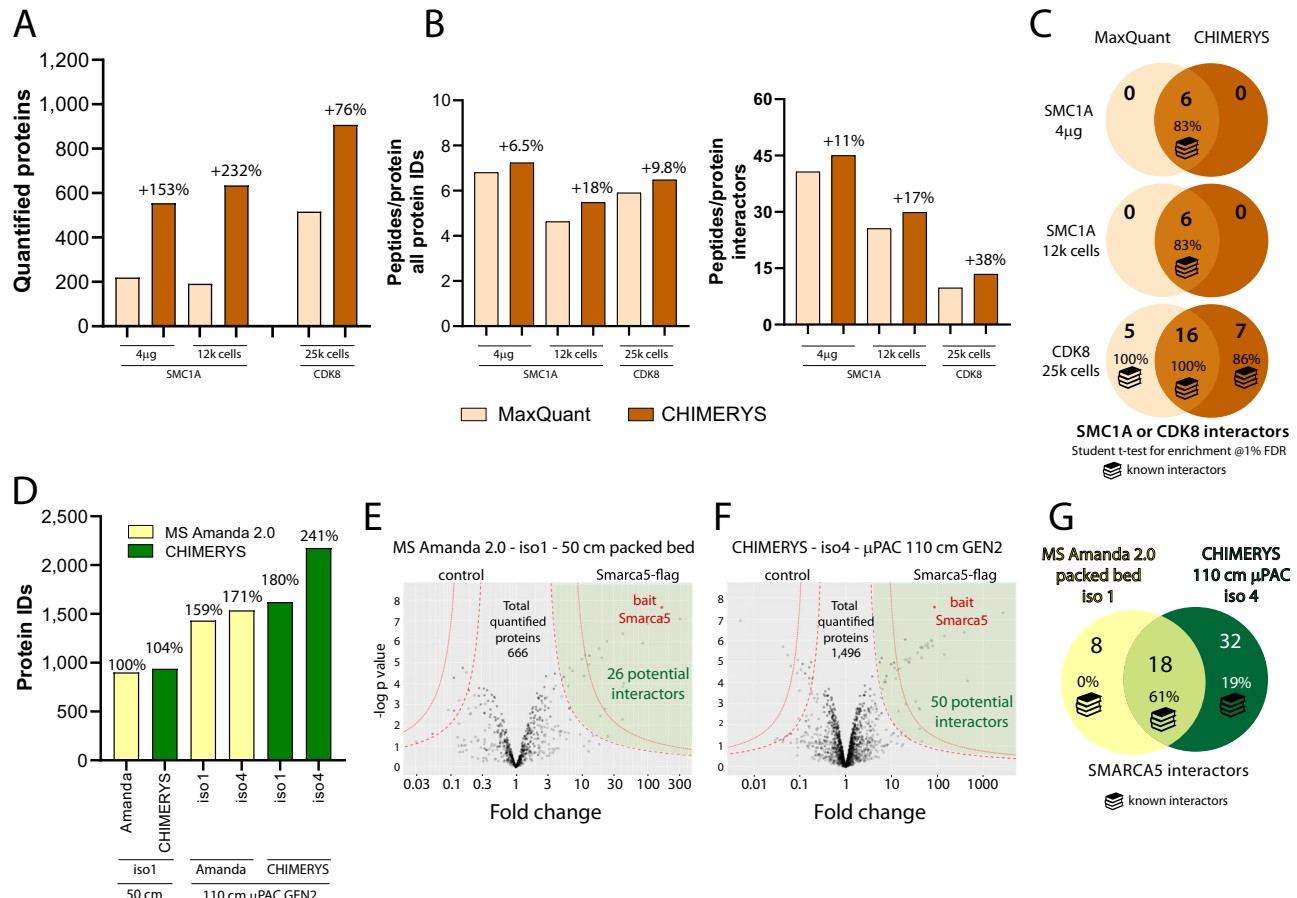

**Fig. 7 | Recapitulation of well-known interactors from previous low input AP-MS data and additional insights into interactions for mouse Smarca5.**
**A–C** Previous low input AP-MS data from ref. 28 was reanalyzed via aforementioned AI-based data analysis and compared to the results reported by Furlan and colleagues. Number of experiments *n* = 1 for all conditions. **A** As compared to the MaxQuant data analysis performed by Furlan et al., CHIMERYS allowed the quantification of 76–232% more proteins for two different baits and three different input amounts (4 μg and 12,000 cells for SMC1A, 25,000 cells for CDK8). **B** Average number of identified peptides/protein is depicted for all proteins (left) and interactors (right), both of which were increased by CHIMERYS for all baits and input amounts (see Supplementary Data 4). **C** Overlay of all interactors identified at 1% FDR for both data analysis platforms indicated near-identical results (see Supplementary Data 4). **D–G** Six mouse co-immunoprecipitation samples were measured

using (i) a 50 cm PepMap column and m/z 1 precursor isolation width, (ii) a μPAC 110 cm column and m/z 1 precursor isolation width, and (iii) a μPAC 110 cm column with m/z 4 precursor isolation width (Supplementary Data 5). **D** demonstrates continuous improvement upon implementation of the different analytical platform constituents. Number of experiments *n* = 1 for all conditions. **E**, **F** display Volcano plots of the basic and the advanced analytic platforms, respectively, based on limma[32] analyses yielding differential enrichment between flag-tagged and WT Smarca5 bait. Finely dashed lines indicate 1% FDR, the lower line indicates 5% FDR. Green area indicates potential Smarca5 interactors. *n* = number of proteins included in the volcano plot, **E** *n* = 666. **F** *n* = 1496. Statistical significance was calculated for both 1% and 5% FDR as described by ref. 44. **G** The advanced analytical platform shows high overlap with the classical approach, but identifies more unique potential interactors. Source data are provided as a Source Data file.

resolution ion trap acquisition and the relatively narrow isolation width of m/z 2 for the reanalyzed data, as CHIMERYS was initially trained for high resolution fragmentation spectra.

To address the impact of our complete optimized platform on the investigation of protein-protein interactions, we analyzed six mouse co-immunoprecipitation samples using the μPAC 110 cm column connected to an Orbitrap Exploris 480 interfaced with a FAIMS Pro device using 4 Th WWA, Proteome Discoverer 3.0, and CHIMERYS for data analysis. The same samples were also analyzed by a more conventional workflow utilizing a 50 cm packed bed column, m/z 1 precursor isolation width and MS Amanda 2.0 for data analysis. The co-immunoprecipitation samples were generated using anti-flag M2 beads specifically enriching for the flag-tagged bait protein. Three samples had been generated using the flag-tagged bait protein, while the other three included non-tagged, wild type bait protein, serving as the control group. The objective of the analysis was the identification of specific interactors to the Smarca5/Snf2h protein in mouse, which is known to be part of numerous complexes involved in chromatin-

remodeling such as the WICH complex or the NoRC-5 ISWI chromatin remodeling complex[29–31].

In line with the other experiments, also substantially more proteins could be identified for the co-immunoprecipitation samples with the advanced analysis platform. While the conventional and the advanced workflow resulted in 901 and 2175 protein IDs, respectively, representing an increase of +141% (see Fig. 7D and Supplementary Data 5). The use of the μPAC column allowed to identify 59% more proteins confirming the high resolving power of μPAC and its applicability for biological projects. Interestingly, the improved chromatography allows CHIMERYS already at an isolation width of m/z 1 to boost IDs more strongly at 21% more proteins when compared to the improvements reached with CHIMERYS for the packed bed column runs with only +4% protein ID increase. As shown earlier in Fig. 5, CHIMERYS reaches its full potential only at higher isolation widths with an ideal isolation width of m/z 4 for regular abundance samples, which again provides an over proportional boost in protein IDs of +61%, when using both μPAC and an isolation width of m/z 4 over m/z 1.

**Table 3 | STRING database analysis of potential interactors identified for Smarca5**

| Included potential interactors | No. of proteins in entire network | Direct Smarca5 interactors | Enriched Gene Ontology terms & pathways | FDR | STRING network |
|---|---|---|---|---|---|
| Unique to classical workflow + Smarca5 | 7 + Smarca5 (1 unmapped protein) | 0 | PPI enrichment p-value 0.0902 -> NOT sign. more interaction partners than expected | |  |
| | | | GO BP – Translation | 0.0239 | |
| | | | GO BP – Cellular nitrogen compound biosynthetic process | 0.0239 | |
| | | | GO BP – Cellular macromolecular biosynthetic process | 0.0239 | |
| | | | KEGG Pathways - Ribosome | 0.0426 | |
| Shared between workflows + Smarca5 | 18 + Smarca5 | 11 | PPI enrichment p < 1.04e-12 -> sign. more interaction partners than expected | |  |
| | | | GO Biological Process - Chromatin remodeling | 3.66E-05 | |
| | | | GO Biological Process - Chromatin assembly or disassembly | 0.0053 | |
| | | | GO Biological Process - Chromatin organization | 0.0053 | |
| | | | GO Cellular Component - ISWI-type complex | 1.13E-05 | |
| | | | GO Cellular Component - NURF complex | 0.0064 | |
| | | | GO Cellular Component - nBAF complex | 0.0378 | |
| | | | GO Cellular Component - npBAF complex | 0.0378 | |
| | | | GO Cellular Component - SWI/SNF superfamily-type complex | 1.13E-05 | |
| Unique to advanced workflow + Smarca5 | 32 + Smarca5 | 6 | PPI enrichment p-value 0.000319 -> sign. more interaction partners than expected | |  |
| | | | GO Biological Process - ATP-dependent chromatin remodeling | 0.0184 | |
| | | | GO Molecular Function - Structural constituent of ribosome | 0.008 | |
| | | | GO Cellular Component - SWI/SNF complex | 0.00056 | |
| | | | GO Cellular Component - npBAF complex | 0.01 | |
| | | | GO Cellular Component - SWI/SNF superfamily-type complex | 0.00075 | |
| | | | GO Cellular Component - Cytosolic small ribosomal subunit | 0.0480 | |
| | | | GO Cellular Component - Cytosolic ribosome | 0.0010 | |
| | | | KEGG Pathways - Hepatocellular carcinoma | 0.0434 | |
| | | | KEGG Pathways – Ribosome | 0.0031 | |

Potential interactors obtained from the Volcano plots (see Fig. 10) were submitted to STRING version 11.5 (https://version-11-5.string-db.org/) in three separate batches: (i) only potential interactors unique to the classical workflow, (ii) potential interactors shared between workflows, and (iii) potential interactors unique to the advanced workflow. Smarca5 was added to each submission as reference point. The network type was set to physical subnetwork and network edges were set to confidence while remaining settings were kept at default and all identified proteins from the respective workflow used as background. While potential interactors specific to the classical workflow did not display significantly more interaction partners than expected, both the shared and the advanced workflow-specific interactors displayed significantly more interaction partners than expected. The advanced platform-specific interactors identified ATP-dependent chromatin remodeling as an additional enriched term while otherwise showing similar enriched GO terms compared to the shared interactors. The full set of enriched terms is listed in Supplementary Data 7.

Furthermore, also substantially more potential interactions as indicated in Fig. 7E, F could be identified. Using the t testing within the statistical package LIMMA[32], 26 and 50 potential interactors could be identified for the conventional and the advanced workflow, respectively, thereby roughly doubling the number of potential interactors. Figure 7G illustrates that the majority of the interactors found by the conventional system could also be reproduced by our advanced platform, of which many are known binding partners of Smarca5 such as Rsf1[33], Bptf[34], Baz2a[29], Cecr2[34], Baz1b[34] and Rbbp4[35]. The eight proteins missed by the advanced platform contained mostly ribosomal proteins, which are frequently occurring, potential contaminants in affinity purification mass spectrometry experiments[36] and did not feature any known Smarca5 interactors. In contrast, the advanced platform-specific 32 interactors include Baz2b, a Smarca5 binding partner within the BRF-5 ISWI chromatin-complex, as well as other proteins indicated in the STRING database to show interaction with Smarca5 such as Smarce1, Actl6a, Bud31 and Nanog[37].

Table 3 also highlights the greater relevance of the potential interactors specific to the advanced platform, with the identification of ATP-dependent chromatin remodeling as enriched biological process and other relevant cellular constituents. In addition, several other potential interactors that were identified with the advanced platform show localization to the nucleus as expected when associated with the chromatin remodeling protein Smarca5 such as Dpf2, Znf280d, Nup62 as well as others, hinting towards a high number of genuine Smarca5 interactors. Arid1a (BAF250a) was also exclusively detected as potential interactor in the advanced platform and has previously been described as component of the SWI/SNF chromatin remodeling complex. Arid1a (BAF250a) is a principal component of SWI/SNF (SWItch/Sucrose Non-Fermentable) family of evolutionary conserved, multi-subunit chromatin remodeling complexes. It interacts with other chromatin remodelers, but to date its direct interaction with Smarca5/ISWI had not been reported in mice to the best of our knowledge. Arid1a holds great clinical relevance as it has been associated with neurodevelopmental[38] and malignant disorders[39,40], and confirmation of direct interaction between Smarca5/ISWI would therefore be of high interest.

## Discussion

In spite of many improvements in mass spectrometry-based proteomics in the last two decades, comprehensive sample throughput and analysis depth are still challenging. The comprehensive analysis of complex proteomic samples up to a depth of 12,000 proteins has been demonstrated by others. However, time consuming offline fractionation was required to achieve this level of comprehensiveness severely hampering sample throughput, which is typically required for large biological or clinical studies and is also of paramount interest for single cell proteomic measurements. In the current manuscript, we describe an advanced workflow to routinely identify more than 10,000 proteins from a triple proteome mix sample of human, yeast and *E. coli* using a 120 min active gradient. In addition, this workflow is also capable of delivering high throughput of up to 96 samples per day when using the 5.5 cm column in combination with short 5 min gradients offering reasonably good proteomic coverage of ~4000 protein IDs for the aforementioned triple proteome mix. We further benchmarked the here used µPAC columns to packed bed columns and found that they deliver significantly more protein and peptide IDs at the same gradient length. We hypothesize that this is due to reduced peak broadening, hence improved separation power and sensitivity. We investigated peak widths obtained with µPAC columns earlier[16] and found a beneficial effect on FWHM that becomes more pronounced with increased gradient length.

We propose an optimized analysis platform combining cutting edge technologies including the Vanquish Neo LC system with micropillar array chromatography, the innovative WWA measurement strategy and the AI-based search engine CHIMERYS. This platform offers high flexibility for the comprehensive and deep analysis of complex samples, or for the effective high throughput analysis of large-scale studies with hundreds of samples in several days. All of the described constituents of the presented advanced analysis platform are commercially available and easily integrated in most proteomics laboratories. Latest generation µPACs for example offer double nanoViper connections facilitating swift installation. The Exploris 480 mass spectrometer as benchtop instrument requires comparatively little space and is easy to work with and also the CHIMERYS software within the Proteome Discoverer framework can be readily applied with little training. In contrast to freely available software packages, however, like MaxQuant, Proteome Discoverer 3.0 and CHIMERYS are only available upon purchase whereas apQuant as quantification node can be downloaded for free. Independently while posterior to a previous preprint version of this manuscript, the Kelly lab likewise describes substantially improved proteome coverage by using WWA in combination with CHIMERYS, corroborating the presented results[24].

Without matching, WWA led to 700 more protein IDs and 470 more protein quantifications at a 2.3% better CV as compared to DDA for 250 pg HeLa bulk samples resembling single cell-like sample inputs. Application of MBR to 10 ng HeLa bulk samples, reduced this beneficial effect of WWA over DDA to afford only 71 additional protein quantifications at a 1.8% better CV. For real single cells, WWA allowed the identification of 415 and the quantification of 244 more proteins at a 7.5% better CV when no matching was applied. Match-between-runs to 40 HeLa cell runs again dampened the ameliorating effect of WWA and only around 60 more proteins could be quantified with WWA over DDA. At around 1000 quantified proteins for a single HeLa cell the presented advanced analytical workflow is on par with other contemporary label-free single cell workflows utilizing similar mass spectrometry technology[26,27,41]. Multiplexed as well as data independent acquisition and time-of-flight-based approaches for single cell proteomics, however, typically yield even improved numbers of protein IDs, while quantitative aspects of these technologies might be a limiting factor for some single cell proteomics projects.

Also, it is demonstrated that this advanced platform recapitulates the vast majority of interactors when applied to the raw data of a previously published low input AP-MS study corroborating the validity of the applied bioinformatic approach. Strikingly, the utilized bioinformatic part of our pipeline resulted in an increased sequence coverage of the identified proteins. This was most pronounced for the two low input samples. Since the reanalyzed data was recorded at low-resolution in an ion trap with a relatively narrow isolation width of m/z 2, these results are all the more encouraging as CHIMERYS has originally been trained on high resolution data, and better performance is to be expected for high resolution data. We therefore hypothesize that the entire analytical workflow presented here will allow to identify substantially more protein-protein interactions for low input AP-MS studies.

Furthermore, it is also illustrated that the entire advanced workflow facilitates the analysis of protein-protein interactions substantially for regular input AP-MS studies by improving the detection of potential Smarca5 interactors by 92% from 26 to 50 as compared to a standard workflow. These additionally identified interaction partners include many known Smarca5 interactors as well as previously unreported ones with clinical relevance such as Arid1a, that has been associated with neurodevelopmental as well as malignant disorders. While Smarca5 is involved in the iswi complex, Arid1a in contrast constitutes the swi/snf chromatin remodeling complex and has previously not been known to directly interact with Smarca5 to the best of our knowledge. This direct interaction observed within this study raises the question whether Arid1a is associated with Smarca5 within the iswi complex or acts as a linker to bridge the iswi complex to the swi/snf complex. In plants it has been described that subunits of iswi complexes can bridge interactions to other chromatin remodeling

complexes like the swr-1 complex[42]. More detailed and focused analyses will be required though in the course of future follow up projects to answer this intriguing question. Overall, the presented workflow offers substantial improvements over current, classical DDA-based proteomic analysis pipelines and will allow to address unanswered questions in biology and the clinics.

## Methods

### Sample preparation for column benchmarking and WWA optimization studies

For initial benchmarking comparing packed columns with μPAC columns, QC4Life Reference standard (CS302403, Promega) was used that consists of LC-MS/MS Peptide Reference Mix (V7491, Promega) in a protein digest of K562 cells (V6951, Promega) diluted to 12.5 ng/5uL with 0.1% TFA.

Triple proteome mixes were made from commercial HeLa (H) (Thermo Scientific, Pierce™ HeLa Protein Digest Standard, 88328), yeast (Y) (Promega, MS Compatible Yeast Protein Extract, Digest, *Saccharomyces cerevisiae*, 100ug, V7461) and *E. coli* digests (E) (Waters, MassPREP *E. coli* Digest Standard, 186003196) combined at a ratio of H:Y:E = 8:1:1 in 0.1% TFA.

Pure HeLa digest samples were prepared from the same HeLa digest as mentioned for the triple proteome mix, which was diluted using 0.1% TFA to reach concentrations of 1 ng/uL for 1 ng injections, 10 ng/uL for 10 ng injections, 200 ng/uL for 100 and 200 ng injections and 800 ng/uL for 400 and 800 ng injections, respectively. To mimic single cell level injections, HeLa digest was diluted to 250 pg/uL in 0.1% TFA including 5% DMSO, and 1uL of this mix was used for injection.

Samples were prepared in glass autosampler vials (Fisherbrand™ 9 mm Short Thread TPX Vial with integrated Glass Micro-Insert; Cat. No. 11515924). All liquid handling was done as fast as possible without unnecessary time gaps aiming to minimize sample adsorption on any surfaces.

### Cultivation of HeLa cells for single cell analysis

Human HeLa cells (ATCC, CCL2) were cultured at 37 °C in a humidified atmosphere at 5% CO₂. HeLa cells were grown in Dulbecco's Modified Eagle's Medium (DMEM) supplemented with 10% FBS (10270, Fisher Scientific, USA), 1x penicillin-streptomycin (P0781-100ML, Sigma Aldrich, Israel) and 100X L-Glut 200 mM (250030-024, Thermo Scientific, Germany). Cells were grown to around 75% confluency before trypsinization with 0.05% Trypsin-EDTA (25300-054, Thermo Scientific, USA), followed by washing 3x with phosphate-buffered-saline (PBS). HeLa cells were resuspended in PBS at a density of 200 cells/μL for isolation with the CellenONE®.

### Double proteome sample preparation

HeLa (H) (Thermo Scientific, Pierce™ HeLa Protein Digest Standard, 88328) and yeast (Y) (Promega, MS Compatible Yeast Protein Extract, Digest, *Saccharomyces cerevisiae*, 100 ug, V7461) were combined in 0.1% TFA at the following ratios in ng/μL: H:Y = 200:0, 150:50, 100:100, 50:150 and 200:0.

### Isolation and sample preparation of single cell and 40 cell samples

HeLa cell isolation, lysis and digestion was performed within a fresh 384 well plate inside the CellenOne® as previously described[25]. Briefly, cells were sorted into well containing 1 μL of master mix containing 0.2% n-dodecyl-beta-maltoside (D4641-500MG, Sigma Aldrich, Germany), 100 mM tetraethylammonium bicarbonate 17902-(500 ML, Fluka Analytical, Switzerland), 3 ng/μL trypsin (Trypsin Gold, V5280, Promega, USA) and 0.01% enhancer (ProteaseMAX, V2071, Promega, USA). For single cell samples, cells were deposited into individual wells, while 40 cells were sorted into a single well for the 40 cell samples. Humidity and

temperature were controlled at 15 °C and 60% during cell sorting. Only HeLa cells with at 18–25 μm diameter and a maximum elongation of 1.5 were isolated. For cell lysis and protein digestion the temperature was increased to 50 °C and the humidity to 85% to limit evaporation. Lysis and digestion were carried out for 2 h at 50 °C at 85% relative humidity inside the instrument. Samples were kept hydrated every 15 min by automated addition of 500 nL water to each well. After 30 min of incubation, additional 500 nL of 3 ng/μL trypsin were added replacing one hydration step. After lysis and digestion, 3.5 μL of 0.1% TFA with 5% DMSO were added to the respective wells for quenching and storage. Samples within the 384 well plates were stored at −70 °C. For the LC-MS/MS analysis, samples were directly injected from the 384 well plate.

### Generation of FLAG tagged Smarca5 cell line

For endogenous tagging of Smarca5, WT mouse ES cells (HA36CB1) grown on a 10 cm plate were transfected with sgRNA/Cas9 ribonucleoprotein complex (sg RNA sequence: TTTGTCTTATAATCACTAAC) and 15 μg of repair plasmid carrying a GFP-3XFLAG tag sequence flanked on both sides by 500 bp of Smarca5 stop codon adjacent homology sequence. For assembling sgRNA/Cas9 ribonucleoprotein complex, 12 ug of sgRNA for Smarca5 was incubated with 5 ug of Cas9 protein in cleavage buffer for 5 min at RT. To transfect sgRNA/Cas9 ribonucleoprotein complex into mouse ES cells, electroporation was carried out following the instructions of the Mouse Embryonic Stem Cell Nucleofector Kit from Lonza (VPH-1001). After 2 days of recovery, GFP expressing cells were FACS sorted (see Supplementary Fig. 7) and seeded for clone picking on 15 cm plate. The clones were individually picked and expanded. Endogenous FLAG tagging of one allele (heterozygous) of *smarca5* was confirmed by genotyping and western analysis.

### Smarca5 Co-Immunoprecipitation

Mouse ES cells were grown on 15 cm plates until confluency. Cells were harvested and washed with 1 × PBS. Then, cells were resuspended in buffer 1 (10 mM Tris-HCl pH 7.5, 2 mM MgCl₂, 3 mM CaCl₂, Protease inhibitors (Roche)) and incubated for 20 min at 4 °C. After centrifugation, cells were resuspended in buffer 2 (10 mM Tris-HCl pH 7.5, 2 mM MgCl₂, 3 mM CaCl₂, Protease inhibitors (Roche), 0.5% IGEPAL CA-630, 10% glycerol) and incubated for 10 min at 4 °C. After this, cells were again centrifuged and nuclei were resuspended in buffer 3 (50 mM HEPES-KOH pH 7.3, 200 mM KCl, 3.2 mM MgCl₂, 0,25% Triton, 0.25% NP-40, 0.1% Na-deoxycholate, 1 mM DDT, Protease inhibitors (Roche). 4 μl of benzonase was added to the nuclei suspension and was incubated for 1 h at 4 °C. The resulting nuclear lysate was cleared by centrifugation. For the Smarca5 IP, WT and Smarca5-Flag were added to magnetic anti FLAG M2 beads (Merck, Sigma-Aldrich, M8823) and incubated at 4 °C for 2 h. Beads were subsequently washed four times with buffer4 (50 mM HEPES-KOH pH 7.3, 200 mM KCl, 3.2 mM MgCl₂, 0,25% Triton, 0.25% NP-40, 0.1% Na-deoxycholate, 1 mM DDT) and four times with Tris buffer (20 mM Tris-HCl pH 7.5, 137 mM NaCl).

### On bead digest of Smarca5 samples

Frozen magnetic beads were thawed, 20 μL of 100 mM ammonium bicarbonate (Merck, Sigma-Aldrich, 09830-1KG) as well as 600 ng of LysC (Wako Chemicals, 129-02541) added and incubated for 4 h at 37 °C at 1200 rpm shaking. Supernatant was aspirated, transferred and tris-(2-carboxyethyl)-phosphin (TCEP, Merck, Sigma-Aldrich, 646547-10X1ML) added up to 1 mM and cysteine reduction performed for 30 min at 60 °C. Reversible blockage of cysteines was performed with S-methyl methanethiosulfonate (MMTS, Merck, Sigma-Aldrich, 64306-1 ML) at 4 mM for 30 min at room temperature. Trypsin digestion was performed overnight with 600 ng trypsin (Promega, V5280) at 37 °C without shaking. Digestion was quenched by addition of 10 μL 10% trifluoroacetic acid (TFA, Thermo Scientific, VC296817).

### Liquid chromatography (LC) and ionization parameters for column benchmarking and WWA optimization studies

All samples were analyzed using a Vanquish Neo UHPLC operated in direct injection mode and coupled to the Orbitrap Exploris 480 mass spectrometer equipped with a FAIMS Pro interface (ThermoFisher Scientific). Analyte separation was performed using prototype versions of either the 5.5 cm High-Throughput µPAC Neo HPLC Column, the 50 cm µPAC Neo HPLC Column or the 110 cm µPAC Neo HPLC Column (all Thermo Fisher) with column volumes of 1.5 µL, 1.5 µL and 4.5 µL, respectively. For benchmarking, classical packed bed columns were used: nanoEase M/Z Peptide CSH C18 Column (130 Å, 1.7 µm, 75 µm X 250 mm, Waters, Germany), PepMap C18 (500 mm × 75 µm ID, 2 µm, 100 Å, Thermo Fisher Scientific) and Aurora Elite G3 (150 mm × 75 µm, 1.7 µm, IonOpticks, Australia). All columns were operated at 50 °C and connected to an EASY-Spray™ bullet emitter (10 µm ID, ES993; Thermo Fisher Scientific) except the Aurora column that already includes an emitter. An electrospray voltage of 2.4 kV was applied at the integrated liquid junction of the EASY-Spray™ emitter for all columns except the Aurora, for which 2.3 kV were used. To avoid electric current from affecting the upstream separation column, a stainless steel 50 µm internal bore reducing union (VICI; C360RU.5S62) was electrically connected to the grounding pin at the pump module for the µPAC columns.

Peptides were separated using gradients ranging from 5 min to 120 min ramping time as detailed in Supplementary Data 1.

### MS Acquisition for column benchmarking and WWA optimization studies

MS acquisition was performed in data-dependent mode, using a full scan with $m/z$ range 380–1200, orbitrap resolution of 60,000, target value 100%, and maximum injection time set to auto. 1 to 4 FAIMS compensation voltages were combined in a single run as detailed in Supplementary Data 6 using a total cycle time of 3 s. The precursor intensity threshold was set to 1e4. Dynamic exclusion duration was based on the length of the LC gradient and is detailed in Supplementary Data 6.

Fragmentation by HCD was done using a normalized collision energy of 30%, and MS-MS spectra were acquired at a resolution of 15,000. Precursors were isolated using a $m/z$ window of 4 for WWA and 1 for normal DDA, respectively, if not stated otherwise.

### Data analysis for column benchmarking and WWA optimization studies

MS/MS spectra from raw data were imported to Proteome Discoverer (PD) (version 3.0.0.757, Thermo Scientific). Database search was performed using MS Amanda[21,22] (version 2.5.0.16129) or CHIMERYS as indicated against a combined database of human (uniprot reference, version 2022-03-04, 20,509 entries), yeast (uniprot reference, version 2015-01-13, 4877 entries) and *E. coli* (uniprot reference, version 2021-11-19, 4350 entries) as well as common contaminants (PD_Contaminants_IGGs_v17_tagsremoved, 344 entries). For HeLa samples, yeast and *E. coli* databases were removed for searches. Trypsin was specified as proteolytic enzyme, cleaving after lysine (K) and arginine I except when followed by proline (P) and up to two missed cleavages were allowed. Fragment mass tolerance was limited to 20 ppm and carbamidomethylation of cysteine (C) was set as a fixed modification and oxidation of methionine (M) as a variable modification. Identified spectra were rescored using Percolator and results were filtered for 1% FDR on peptide and protein level. Abundance of identified peptides was determined by label-free quantification (LFQ) using IMP-apQuant without MBR[43].

### LC-MS/MS analysis of single and 40 cell samples

All single and 40 cell samples were analyzed using a Vanquish Neo UHPLC operated in trap-and-elute mode and coupled to the Orbitrap Exploris 480 mass spectrometer equipped with a FAIMS Pro interface (ThermoFisher Scientific). Peptides were loaded on a trapping column (Thermo Fisher Scientific, PepMap C18, 5 mm × 300 µm i.d., 5 µm particles, 100 Å pore size) using 0.1% TFA as the mobile phase. Peptides were eluted from the trapping column onto a prototype versions of the 5.5 cm High-Throughput µPAC Neo HPLC Column (Thermo Fisher Scientific) using a flow rate of 250 nl/min with a gradient length of 20 min. The gradient started with mobile phases of 99% A (water:formic acid, 99.9:0.1 v/v) and 2% B (water:acetonitrile:formic acid, 19.92:80:0.08 v/v/v), increasing first to 25% B in 14.5 min before ramping up to 40% B in 5 min, followed by a gradient over 5 min to 95% B, that was held for 5 min and decreasing in 0.1 min back to 99% A and 1% B for equilibration at 50 °C.

The Orbitrap Exploris 480 mass spectrometer was operated in the data-dependent mode with the FAIMS Pro using a single compensation voltage of −50 V. A full scan (m/z range of 375–1200, MS1 resolution of 120,000, normalized AGC Target of 300%) was followed by up to 10 MS/MS scans of the most abundant ions. MS/MS spectra were acquired using a normalized collision energy of 30%, an isolation width of m/z 1 for DDA runs and m/z 12 for WWA runs and a resolution of 60,000 with a normalized AGC target of 75%. Precursor ions selected for fragmentation (exclude charge state 1, 6, 7, 8 and >8) were placed on a dynamic exclusion list for 120 s. Additionally, the intensity threshold was set to a minimum intensity of $5 \times 10^3$.

### Single HeLa cell data analysis

MS/MS spectra from raw data were imported to Proteome Discoverer (PD) (version 3.0.0.757, Thermo Scientific). Database search was performed using CHIMERYS as indicated against a human database (uniprot reference, version 2022-03-04, 20,509 entries) as well as common contaminants (PD_Contaminants_IGGs_v17_tagsremoved, 344 entries). Trypsin was specified as proteolytic enzyme, cleaving after lysine (K) and arginine (R) except when followed by proline (P) and up to two missed cleavages were allowed. Carbamidomethylation of cysteine (C) was set as a fixed modification and oxidation of methionine (M) as a variable modification. Identified spectra were rescored using Percolator and results were filtered for 1% FDR on peptide and protein level. Abundance of identified peptides was determined by label-free quantification (LFQ) using IMP-apQuant without MBR[43].

### LC-MS analysis of immunoprecipitation samples

The nano HPLC system used was an UltiMate 3000 RSLC nano system coupled to a Orbitrap Exploris 480 mass spectrometer, equipped with an EASY-spray ion source (Thermo Fisher Scientific) and a JailBreak 1.0 adaptor insert as the spray emitter (Phoenix S&T) as well as a FAIMS Pro device (Thermo Scientific). Peptides were loaded on a trapping column (Thermo Fisher Scientific, PepMap C18, 5 mm × 300 µm i.d., 5 µm particles, 100 Å pore size) at a flow rate of 25 µl/min using 0.1% TFA as the mobile phase. 10 min after sample injection, the trapping column was switched in line with the analytical column and peptides were eluted from the trapping column onto the analytical column using a flow rate of 230 nl/min and a binary 2 h gradient was employed. MS acquisition was started 10 min after switching the trap column in line with the analytical column for a total MS acquisition time of 140 min, a total run length of 165 min and 120 min active gradient. The analytical column was either a classical 50 cm packed bed column (Thermo Fisher Scientific, PepMap C18, 500 mm × 75 µm i.d., 2 µm, 100 Å) or a prototype version of the 110 cm µPAC Neo HPLC Column (Thermo Fisher Scientific, micropillar array column, C18) The gradient started with mobile phases of 98% A (water:formic acid, 99.9:0.1 v/v) and 2% B (water:acetonitrile:formic acid, 19.92:80:0.08 v/v/v), increasing to 35% B over the next 120 min, followed by a gradient over 5 min to 95% B, held for 5 min and decreasing over 2 min back to gradient 98% A and 2% B for equilibration at 30 °C. The trapping column was switched out of line from the analytical column 3 min after reaching 2% B again, and equilibration at 2% B was continued until the total run time of 165 min was reached.

The Orbitrap Exploris 480 mass spectrometer was operated in the data-dependent mode with the FAIMS Pro using three different compensation voltages at −45, −60 and −75 in an alternating fashion switching between compensating voltages every 0.9 s. A full scan (m/z range of 350–1200, MS1 resolution of 60,000, normalized AGC Target of 100%) was followed by MS/MS scans of the most abundant ions until the cycle time of 0.9 s was reached. MS/MS spectra were acquired using a normalized collision energy of 30%, an isolation width of m/z 1, a resolution of 30,000 and a normalized AGC target of 200%. Precursor ions selected for fragmentation (exclude charge state 1, 7, 8 and >8) were placed on a dynamic exclusion list for 45 s. Additionally, the intensity threshold was set to a minimum intensity of $2.5 \times 10^4$.

### Data analysis of immunoprecipitation raw data

For peptide identification, RAW files were loaded into Proteome Discoverer (v.3.0.0.757, Thermo Scientific). All the created MS/MS spectra were searched either using MSAmanda v.2.0[21,22] or CHIMERYS (MSAID GmbH, Germany). For the processing step, the RAW files were searched against the mouse Uniprot reference database (2022-03-04; 21,962 sequences and 11,728,099 residues) and an in-house contaminant database (PD-Contaminants_IGGs_v17_tagsremoved; 344 sequences and 142,046 residues).

The following search parameters were used for MS Amanda 2.0: the peptide mass tolerance was set to ±5 ppm and the fragment mass tolerance to 10 ppm; the maximal number of missed cleavages was set to 2; and the result was filtered to 1% false discovery rate (FDR) on the protein level using the Percolator algorithm integrated in Thermo Proteome Discoverer. Beta-methylthiolation on cysteines was set as fixed modification, whereas methionine oxidation was set as variable modification.

For CHIMERYS the following search parameters were used: as prediction model inferys_2.1_fragmentation was chosen using trypsin with a maximum of two missed cleavages. Peptide length was restricted to 7–30 amino acids, a maximum of three modifications per peptide and a charge state of 2–4. Fragment mass tolerance was set to 20 ppm. Methionine oxidation was set as variable modification, while cysteine carbamidomethylation was predefined by the software as fixed modification and could not be unselected even though the samples had been treated with methyl methanethiosulfonate (MMTS) to reversibly sulfenylate cysteine introducing beta-methylthiolation. However, since only around 8% of the peptides identified with MS Amanda 2.0 contained cysteines, this incorrect parameter was considered negligible in the course of this analysis, while using the correct fixed modification would have surely resulted in even better results for all CHIMERYS searches.

Peptide areas were quantified using IMP-apQuant[43] using only PSM of high confidence level, with a minimum sequence length of 7 and a minimum score of 150 for MS Amanda 2.0 and −99 for the CHIMERYS Ion Coefficient and match-between-runs and RT correction were disabled. Retention time tolerance was set to 0.5, missing peaks to 2 and FWHM interpolation was enabled, and number of checked peaks set to 5. The results were filtered to 1% FDR on the protein level using the Percolator algorithm integrated in Thermo Proteome Discoverer. Statistical significance of differentially abundant peptides and proteins between different conditions was determined using a limma test[32]. Statistical significance was calculated for both 1% and 5% FDR as described by ref. 44

### Reporting summary

Further information on research design is available in the Nature Portfolio Reporting Summary linked to this article.

## Data availability

The mass spectrometry proteomics data have been deposited to the ProteomeXchange Consortium via the PRIDE [131] partner repository with the dataset identifiers PXD037985 (for Smarca5 AP-MS data), PXD039576 (for μPAC benchmarking data), PXD045457 (for Aurora column data) and PXD045500 (for all remaining data). For benchmarking studies, data from ref. 28 were downloaded from the ProteomeXchange server using the accession number PXD012800. STRING database version 11.5 was used to create all networks containing 67,592,464 proteins from 14,094 organisms with 20,052,394,042 interactions, which can be accessed via https://version-11-5.string-db.org/. Source data are provided with this paper.

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

## Acknowledgements

This work supported by the EPIC-XS, Project Number 823839 (K.M., Grant-DOI 10.3030/823839), funded by the Horizon 2020 Program of the European Union (K.M.), by the project LS20-079 of the Vienna Science and Technology Fund (K.M.) and the by the by ERA-CAPS I 3686-B25-MEIOREC (K.M.), P35045-B (K.M., Grant-DOI 10.55776/P35045), P32054 (F.B., GrantDOI 10.55776/P32054) and P33380 (F.B., Grant-DOI 10.55776/P33380) projects of the Austrian Science Fund. We thank the IMP, IMBA and GMI for general funding and access to infrastructure and especially the technicians of the protein chemistry facility for continuous technical support. All LC–MS/MS analyses were performed on instruments of the Vienna BioCenter Core Facilities instrument pool. The mouse embryonic stem cells were kindly provided by Julius Brenneke. We are grateful to MSAID and Thermo Fisher Scientific in particular to Bernard Delanghe, Martin Frejno and Daniel Zolg for the opportunity to test CHIMERYS and Proteome Discoverer 3.0. We also thank Thermo Fisher Scientific for access to the micropillar array columns, especially Jeff Op de Beeck and Paul Jacobs.

## Author contributions

RM and MM conceptualized the study, performed experiments, data analysis and wrote the manuscript. RM and MM contributed equally to this work. AS and RY prepared Smarca5 IP samples. GK and KS maintained, and reorganized LC-MS systems used for this work. FB conceptualized work on Smarca5. KM conceptualized the study and performed data analysis.

## Competing interests

The authors declare no competing interests.

## Additional information

[1]Research Institute of Molecular Pathology (IMP), Vienna BioCenter, Vienna, Austria. [2]Gregor Mendel Institute of Molecular Plant Biology (GMI), Austrian Academy of Sciences, Vienna BioCenter (VBC), Vienna, Austria. [3]MRC (Medical Research Council) London Institute of Medical Sciences, Du Cane Road, London W12 0NN, UK. [4]Institute of Clinical Sciences, Imperial College London, Hammersmith Hospital Campus, Du Cane Road, London W12 0NN, UK. [5]Institute of Molecular Biotechnology (IMBA), Austrian Academy of Sciences, Vienna BioCenter (VBC), Vienna, Austria. [6]Laboratory of Epigenetics, Cell Fate & Disease, Centre for DNA Fingerprinting and Diagnostics (CDFD), Uppal, Hyderabad, India. [7]These authors contributed equally: Manuel Matzinger, Rupert L. Mayer. ✉e-mail: manuel.matzinger@imp.ac.at; karl.mechtler@imp.ac.at; rupert.mayer@imp.ac.at

