## [Peer Review File · Nature Communications]

Micropillar arrays, wide window acquisition and AI-based data analysis improve comprehensiveness in multiple proteomic applicationsREVIEWER COMMENTS

Reviewer #1 (Remarks to the Author):

In their manuscript, Mayer & Matzinger et al. present an elaborate evaluation of a novel multi-component LC-MS pipeline, consisting of new chromatography platforms (uPAC), a novel data acquisition scheme (wide-window acquisition, WWA) and very recent data analysis software (Chimerys). Overall, the work is well conducted and of high quality, and should appeal to a broad readership interested in the latest state-of-the-art within biological LC-MS based proteomics. The authors also chose to evaluate their analytical pipeline in a range of applications (bulk MS, affinity-purification MS and single-cell proteomics), which adds to the overall impact and broad applicability of their work.

In general, the manuscript is well written with a solid experimental design, delivering elaborate insights into the various aspects affecting the performance of their pipeline (i.e. column type, gradient length and sample load). The reader is provided with an overview of which options to implement in their own studies, and I expect these results to transfer well to other labs. Especially with the referred-to “prototype column” now having been released to the market, all aspects of the presented work are universally implementable, making these results very relevant for the proteomics field. The authors are also very successful to clearly convey results with many variables involved, elegantly depicting them within the figures and making them easy to follow - they should be commended for this.

The manuscript in my view is not far from being ready for publication, but I would like to request clarifications / corrections on the (mainly minor) items listed below:

1. I wonder if the authors conducted some kind of injection time (IT) survey on the various load? I would expect substantially different settings in that regard to be optimal on high vs low load? Also in light of the substantial dip in performance from 50ng to 10ng, wouldn't this gap disappear / be alleviated if IT was increased for the <50ng injections?

2. A re-appearing discussion in the field of single-cell proteomics is whether 250pg accurately mimics single-cell - can their data corroborate this or should even lower amounts

(dilution) be used to better reflect sample losses involved with single-cell prep?

3. According to the methods, Max IT on MS1 scans was set to auto; what does that mean in practise for the MS1 scan ITs, how long were average injection times and would hardcoding this to be parallel with the Orbitrap transient produce better results in low load? A brief discussion of this would be nice.

4. Regarding their CellenOne-based cell sorting, why were only 18-25um cells sorted? What was the overall cell distribution and do the sorted cells represent average, small or large sizes?

5. For the single-cell-proteomics introduction, I would argue that some key references are missing (Specht et al. 2021, Budnik et al. 2018, Schoof et al. 2021, Zhu et al. 2018, Ctortecka et al, BioRxiv 2021) to allow placing this work into their proper context. Also in terms of DIA, for completeness sake I think Brunner et al., 2021 and Petrosius et al., BioRxiv 2022 should be listed as well.

6. Please update the brick prototype column to the now-released version which I believe was launched at ASMS. I also believe the 110cm one referred to as prototype on line 127 is now a standard GEN2 column (e.g. line 149)?

7. Lines 288-290 seem to be very well in line with wide-window DIA as presented in Petrosius et al., BioRxiv 2023. It would be nice if the authors could refer to this work as both sets of results corroborate each other.

8. Regarding the MBR+ vs MBR- results, it would be nice to see how the CVs are different for those proteins measured without MBR from those proteins that were added through the MBR. I would expect the latter set (ie. MBR-only proteins) to be much more noisy in terms of their reproducibility, especially on the low / single-cell loads, which would be a critical point to evaluate.

9. Regarding the gradient design, the reader is left wondering why the washing time is

dependent on the gradient length? Should it not rather be dependent on the column volume? E.g. washing for 1 column volume? They also do not mention the column volume specification, which is an important parameter for gradient design. E.g. for the 30min gradient, all 3 columns are washed for 15min at 300nl/min, but I suppose that the column volume is different between these 3 columns?

Similar questions arise for the equilibration? How many column volumes are needed at which flow rate? And for the loading, how much volume and at which flow rate?

Lastly, how does the 47min overhead for the 110cm column result? At a flow rate of e.g. 500ul/min, that would be a total of 23ul used for loading and equilibration. Why is that needed?

10. On lines 488 & 491 it should be clarified that this is a human:yeast:ecoli proteome mix.

11. A few more details should be provided in the conclusion/discussion on why the uPAC column outperform the packed columns at the same gradient length? (I guess peak shape/widths?)

12. Could the authors comment on whether the additional IDs from WWA impact on better quantification of proteins compared to standard DDA, ideally both in terms of accuracy and precision? Also, how many Chimerys-derived PSMs are actually linked to MS1 features?

Reviewer #2 (Remarks to the Author):

Thank you for the opportunity to read and review "Micropillar array chromatography, wide window acquisition and AI2 based data analysis enable deep coverage in a broad range of proteomic applications from complex mixtures to limited amount samples".

As the title suggests, there is a lot of data in this manuscript. While that is indicative of the amount of work that is summarized here, it also makes the work, as described, challenging to follow. Highlights for the study include this being one of the first few descriptions of the Chimerys search engine, In addition, there has been relatively little work reported on the

openly available IMP-APQuant proteomics quantification algorithm. I find it impressive that the authors are comfortable not only comparing the commercial Chymerys engine to their own well-established search engine MS Amanda, despite the fact that Chymerys appears to outperform their own tools. Given the performance values here it seems that the authors are suggesting the use of the commercial Chymerys engine for spectral library assignment and the use of the freely available APQuant algorithm is an ideal combination. It would be nice to see a note in the text on whether this combination is possible to obtain without purchasing the complete commercial solution from Thermo Fisher. If so, this could be a very powerful option for many proteomics labs at a substantially reduced cost.

Major comments:

The work would benefit from a main text table or schematic that summarizes or provides structure to the work that is described in the text. The very large Excel file (which I believe is Supplemental Data 1 - but the identifiers provided by SpringerLink make difficult for this reviewer to follow) which summarizes 40 cell, single cell and 250 pg peptide load data itself is a very interesting story of optimization for low input reproducible proteomics. The immunoprecipitations provide evidence that this commercial cloud based search engine adds additional biological insight. Again, very interesting, but in this current state this reads more like 3 stories rather than a single cohesive manuscript.

The data deposited on ProteomeXchange seems to be complete and fully support the findings as described. Each piece of the manuscript is well written.

With something to help clarify and bring structure to the work described here, I'd consider this a work suitable for publication in this journal.

Minor comments

Line 688 methods: Please clarify the gradient length vs total acquisition time. I think I see what you mean, but I also think it would be helpful for other readers if this was better explained

Figure 7: I don't think there is enough value in Figure 7 for inclusion as a main text figure. I'd suggest moving it to supplemental or inclusion as statements in the text alone.

Figure 8: I suspect this figure is too busy to be displayed appropriately in a publication, particularly with text within the Venn diagrams in Fig 8C.

Supplemental Data 2 (summary of immunoaffinity results) tabs containing Proteome Discoverer output tables could be simplified to remove data not useful to readers. Whether or not the protein was "checked" prior to exporting should be deleted.

Reviewer #3 (Remarks to the Author):

Summary: The authors present the combination of micropillar array chromatography with wide window analysis and a machine learning based search algorithm to demonstrate deep coverage of low level samples. The authors first comparing packed bed and micropillar columns before evaluating different formats of micropillar columns. The authors then explore various gradient lengths to demonstrate the potential impact of a shorter micropillar column. After choosing the optimal column – the authors compare a machine learning algorithm (CHIMERYS) against a more traditional search engine (MS Amanda 2.0) to demonstrate that CHIMERYS returns more identifications, particularly when using a wide window acquisition technique. The authors then analyze various low level samples and AP-MS experiments to demonstrate the boost in sensitivity of their approach.

The manuscript presents data for a number of different optimizations within their workflow to present an approach that appears to work well for low-level samples. This is a timely topic within the proteomics field as there are a number of new columns, data acquisition techniques, and data analysis approaches that are currently being introduced. One major point I would raise would be the reporting of identified proteins, rather than quantified proteins. For the application of this approach it is important to understand how many proteins can be accurately quantified – rather than just identified. Furthermore, I found figures 3 and 5 to be the most impactful as they provide a well-controlled comparison of the various columns, gradient lengths, and data acquisition strategies. Other figures (e.g., 2, 4, 6) provide intriguing data, but could be expanded upon based on the questions below. Overall, this manuscript would be of interest to the field and the readers of Nature Communications and in my opinion would be suitable for publication following revisions that address questions below.

Major Points:

1. Protein identification vs. quantification: The figures within the manuscript report the number of identified proteins. However, evaluation of this (and all methods) should comment on the number of proteins that can be accurately quantified as well. In order to utilize this approach to answer meaningful biological questions it will be important to demonstrate that the WWA and CHIMERYS searching algorithm returns additional quantified proteins in addition to identifying more proteins.

2. Figure 2. Comparison to non-micropillar column: For the comparison of packed bed and micropillar columns, the variety of packed bed columns could be improved. Specifically, many researchers within the low-level and single cell proteomics field are using IonOpticks columns for their analysis. IonOpticks columns are known to have very high performance, so it would be good to demonstrate the comparison to these columns within this figure.

3. Figure 5. Optimal window size increases for lower level samples: The authors demonstrate that with decreasing amount of peptide analyzed the optimal WWA window increases. Do the authors believe that this is due gas phase enrichment of precursors that were not detected and triggered on in lower level samples? How do the number of MS/MS events compare between these analyses – are more MS/MS being taken with higher peptides loads? Since WWA depends on the identification of a peak in the MS1 it would stand to reason that smaller windows may not perform well in lower level samples because there are not enough peaks to trigger WWA on a wide region of the precursor m/z range.

4. Figure 6. WWA improves CVs: Similarly, the authors state that WWA improves CVs though lower injections times. Can the authors demonstrate the number of points across the curve for quantified peptides to demonstrate that WWA changes the number of data points collected – potentially leading to an improvement in quantitation? Is it possible that the WWA doesn't increase the number of the points across the curve, but rather allows peptides to be samples closer to their apex through repeated sampling of particular m/z ranges?

5. Figure 7. Broader dynamic range with WWA: This figure does not provide sufficient evidence that the dynamic range is significantly different with WWA. I would either remove

this figure or collect additional data to demonstrate that the dynamic range is significantly different (with replicates, etc.).

Minor Points:

1. Figure 4. Proteins per minute: This may be a personal preference – but protein IDs per minute of total runtime is not too useful of a metric for those looking to understand how many samples could be run in a day while maintaining a certain proteome depth. It would be interesting to see these data plotted as the number of samples that could be analyzed per day with a depth of X proteins (e.g., 4,000 or 6,000 proteins). This would help to demonstrate the differences in the throughput without readers needing to do the math in their head.

REVIEWER COMMENTS

Reviewer #1 (Remarks to the Author):

In their manuscript, Mayer & Matzinger et al. present an elaborate evaluation of a novel multi-component LC-MS pipeline, consisting of new chromatography platforms (uPAC), a novel data acquisition scheme (wide-window acquisition, WWA) and very recent data analysis software (Chimerys). Overall, the work is well conducted and of high quality, and should appeal to a broad readership interested in the latest state-of-the-art within biological LC-MS based proteomics. The authors also chose to evaluate their analytical pipeline in a range of applications (bulk MS, affinity-purification MS and single-cell proteomics), which adds to the overall impact and broad applicability of their work.

In general, the manuscript is well written with a solid experimental design, delivering elaborate insights into the various aspects affecting the performance of their pipeline (i.e. column type, gradient length and sample load). The reader is provided with an overview of which options to implement in their own studies, and I expect these results to transfer well to other labs. Especially with the referred-to "prototype column" now having been released to the market, all aspects of the presented work are universally implementable, making these results very relevant for the proteomics field. The authors are also very successful to clearly convey results with many variables involved, elegantly depicting them within the figures and making them easy to follow - they should be commended for this.

The manuscript in my view is not far from being ready for publication, but I would like to request clarifications / corrections on the (mainly minor) items listed below:

We are highly grateful to the reviewer for his kind support and his/her comprehensive and helpful assessment of our manuscript. The authors highly appreciate the substantial amount of time and effort the reviewer has clearly invested to help us improve our manuscript. We hope that our replies and further efforts to improve our work will justify publication of this manuscript in Nature Communications in the eyes of this reviewer.

1. I wonder if the authors conducted some kind of injection time (IT) survey on the various load? I would expect substantially different settings in that regard to be optimal on high vs low load? Also in light of the substantial dip in performance from 50ng to 10ng, wouldn't this gap disappear / be alleviated if IT was increased for the <50ng injections?

We thank the reviewer for this valuable comment. Indeed, we also expect considerable differences between the optimal injection times for low inputs like 10ng or 50ng versus regular inputs like 400ng.

The reviewer's question sparked us to conduct a short survey (see Reviewer Figure 1), which confirms the expected differences in ideal maximum filling times clearly showing that particularly the 10ng input amount would have benefitted from longer filling times.

We have added a new Supplemental Figure 1 and included the following sentence in the main text to: "The striking protein ID gap between an injection amount of 10 ng and 50 ng had not been

reported by earlier studies in our lab, but might largely be attributed to the rather short MS2 maximum injection times of 23 ms used here as demonstrated in supplemental Figure 1”

In addition, the ID gap may be explained by the lower concentration of the 10ng sample. While the 50ng injection was aspirated from a 100ng/ μ L sample, the 10ng injections were aspirated from 10ng/ μ L samples. Even though the samples were prepared only a few hours prior to the actual measurements, sample losses due to adsorption on the tube wall might be more pronounced for the 10ng input and contribute to the observed phenomenon.

2. A re-appearing discussion in the field of single-cell proteomics is whether 250pg accurately mimics single-cell - can their data corroborate this or should even lower amounts (dilution) be used to better reflect sample losses involved with single-cell prep?

We thank the reviewer for this important question and agree this is indeed a re-appearing discussion in the field of single cell proteomics. That said, the authors believe that there is no single protein amount that correctly reflects the actual content of a real single cell. The protein content will likely depend on cell-size, cell cycle stage, cellular stress, density while cultured, if fresh media as recently added and other parameters. Volpe et al. investigated the protein content of HeLa cells already back in 1970 (<https://doi.org/10.1111/j.1432-1033.1970.tb00837.x>) and showed the huge variances possible. Based on their results 250 pg seems a reasonable value. In the Mechtler lab, we decided for this amount to represent ideal conditions in case of ‘perfect’ sample preparation. Compared to real HeLa cells, in our hands, 250 pg QC runs correlate to the best-case scenario. To maintain consistency and comparability also with our previous works (e.g. Stejskal et al. <https://doi.org/10.1021/acs.analchem.1c00990>, Matzinger & Müller et al. <https://doi.org/10.1021/acs.analchem.2c05022>) we used that very same concentration in this study. Of course, lower concentrations could be included as dilution series, however, we do not expect significant novel findings from that as our existent dilution series (Figure 5D,E) already shows a clear trend to wider windows being advantageous with reduced input amount. Real single cells in comparison to our 250 pg stock are included in Figure 6, allowing to compare differences from the chosen stock to real cells in terms of protein numbers identified, quantified, and CVs.

3. According to the methods, Max IT on MS1 scans was set to auto; what does that mean in practise for the MS1 scan ITs, how long were average injection times and would hardcoding this to be parallel with the Orbitrap transient produce better results in low load? A brief discussion of this would be nice.

When “Auto” is selected the maximum ion injection time is aligned with the Orbitrap transient time allowing a large fraction of the Orbitrap transient time n ($n=9$ ms for 15k-480k resolution and $n=4$ ms for 7.5k resolution) to be utilized for ion accumulation in the ion-routing multipole.

For testing the different μ PAC columns, the MS1 resolution was set to 60,000 resulting in an Orbitrap transient time of 128ms with a parallelizable time and maximum MS1 IT of 119ms. For the analysis of the low input samples, the MS1 resolution was set to 120,000 as longer ion injection times were

expected to be necessary anyhow, resulting in an Orbitrap transient time of 256ms with a parallelizable time and maximum MS1 IT of 247ms.

Average MS1 injection times during the active peptide elution for higher inputs were 6.13ms for 10ng, 2.15ms for 50ng, 1.57ms for 100ng, 0.45ms for 200ng and 0.20ms for 400ng of HeLa:yeast:E.coli using a 30 min gradient method and the 50cm μ PAC, which is far below the maximum set injection time of 119ms for MS1 and 23ms for MS2 and allows efficient parallelization with the last MS2 scan of the previous duty cycle. For the low load samples including the single cell and 250pg HeLa samples, the MS1 injection times for our chosen method with a 20min gradient was on average around 82ms. This again matches nicely the maximum MS2 transient time for the same method with 119ms. We therefore believe that no large gains in IDs are to be expected for both methods when adapting the MS1 injection time.

4. Regarding their CellenOne-based cell sorting, why were only 18-25 μ m cells sorted? What was the overall cell distribution and do the sorted cells represent average, small or large sizes?

In our manuscript we direct the interested reader to a previous work of our group (Matzinger & Müller et al. <https://doi.org/10.1021/acs.analchem.2c05022>), where sample preparation is described. There we found for our HeLa cells that the majority of cells had diameters ranging from 15 to 30 μ m, which is in line with results from others (Molly et al <https://doi.org/10.1006/mthe.2000.0054>). The selected size-range therefore represents average-sized cells. As described in Matzinger & Müller et al., we optimized cell isolation parameters aiming for isolating exactly one viable cell in a fully automated manner and this is why the diameter was chosen accordingly. With less stringent windows the error rate of the CellenONE increased in our hands leading to wells with no intact cell, cell debris or two cells isolated.

5. For the single-cell-proteomics introduction, I would argue that some key references are missing (Specht et al. 2021, Budnik et al. 2018, Schoof et al. 2021, Zhu et al. 2018, Ctortecka et al, BioRxiv 2021) to allow placing this work into their proper context. Also in terms of DIA, for completeness sake I think Brunner et al., 2021 and Petrosius et al., BioRxiv 2022 should be listed as well.

We thank the reviewer for this valuable suggestion. This study focuses on WWA as acquisition method in combination with a μ -pillar columns and a novel data analysis software (Chimerys), which is why also the introduction focuses on those topics. However, we agree that since also single cells were measured for this manuscript, it's important to reference the aforementioned single cell works to provide context to the figure containing single cell data within this study. We included a brief statement in the introduction of the revised manuscript and added the listed literature.

6. Please update the brick prototype column to the now-released version which I believe was launched at ASMS. I also believe the 110cm one referred to as prototype on line 127 is now a standard GEN2 column (e.g. line 149)?

We are grateful to the reviewer for pointing this out. We have changed the names of the columns to make the reader aware that updated versions of the columns used in this paper are commercially available even though some characteristics and limitations of the commercial columns slightly differ from the early column versions we obtained for testing. This should allow readers to find the respective commercial columns with ease and facilitate implementation of the proposed workflow.

7. Lines 288-290 seem to be very well in line with wide-window DIA as presented in Petrosius et al., BioRxiv 2023. It would be nice if the authors could refer to this work as both sets of results corroborate each other.

We thank the reviewer for this valuable suggestion and agree that the work of Petrosius et al. done on a DIA strategy is very well in line with our observations using WWA. We added a respective note in the revised manuscript and cited their work.

8. Regarding the MBR+ vs MBR- results, it would be nice to see how the CVs are different for those proteins measured without MBR from those proteins that were added through the MBR. I would expect the latter set (ie. MBR-only proteins) to be much more noisy in terms of their reproducibility, especially on the low / single-cell loads, which would be a critical point to evaluate.

The reviewer's request sparked our own curiosity and prompted us to investigate the CVs of MBR-only proteins in all 250pg, single cell, 10 ng as well as 40 cell samples. Interestingly, the CVs of the MBRonly proteins were only worse for the single cell samples, but markedly better than the proteins quantified without MBR for both 250 pg and 10 ng (see Reviewer Figure 1). The 40 cell samples displayed similar, but slightly improved CVs for the MBRonly proteins. While we also indeed expected worse CVs for the MBRonly proteins, the observed effect for 250 pg and 10 ng might be explained by more stable and reproducible peak picking and signal integration due to the matching. We also have to add here that during the revision process an inconsistency in the assessment of the CVs for single and 40 cells without MBR has been corrected. Therefore, CVs between for WWA are only improved consistently over DDA when MBR is applied. We have assessed in Supplemental Figure 5 if the improved quantification for WWA when applying FDR arises from additional points over the peak, but DDA and WWA runs demonstrated identical median datapoints per peak. The authors therefore believe that additional peptides identified and quantified when applying WWA could stabilize protein intensities which takes effect only upon cross-run normalization and matching as observed in the MBR analysis.

Reviewer Figure 1: **Comparison of median CVs for noMBR and MBRonly proteins.** Coefficients of variation for proteins quantified in all 3 replicates without match-between-runs were compared to those proteins quantified in all three replicates exclusively when using MBR. These proteins include all proteins that were quantified in two or fewer replicates of the samples analyzed without MBR and in all three replicates of the samples analyzed with MBR.

9. Regarding the gradient design, the reader is left wondering why the washing time is dependent on the gradient length? Should it not rather be dependent on the column volume? E.g. washing for 1 column volume? They also do not mention the column volume specification, which is an important parameter for gradient design. E.g. for the 30min gradient, all 3 columns are washed for 15min at 300nl/min, but I suppose that the column volume is different between these 3 columns?

Similar questions arise for the equilibration? How many column volumes are needed at which flow rate? And for the loading, how much volume and at which flow rate?

Lastly, how does the 47min overhead for the 110cm column result? At a flow rate of e.g. 500ul/min, that would be a total of 23ul used for loading and equilibration. Why is that needed?

We thank the reviewer for mentioning this point. We agree that the duration of the washing phase should be mainly dictated by the column volume but will also be heavily influenced by the sample type loaded onto the column. Digested blood plasma or plant samples for example will mandate more extensive washing of the chromatographic system than a simple purified cell digest in the experience of the authors. In our case, the estimated degree of washing was deemed moderate since the samples consisted of cellular lysates of HeLa, E.coli and yeast.

As the reviewer pointed out, the column volumes for the 5.5 cm and 50 cm columns (both 1.5 μ L) are indeed different to the 110cm column (4.5 μ L). All column volumes are publicly available (<https://www.thermofisher.com/document-connect/document-connect.html?url=https://assets.thermofisher.com/TFS-Assets%2FCMD%2Fmanuals%2Fxx-001891-ccs-upac-neo-hplc-columns-start-up-xx001891-en.pdf>) and we have also now added this information in the results part as well as the materials and methods part. As indicated by the reviewer, these different column volumes, while using the same flow rate and time for washing, resulted in different column volumes of washing for the 110 cm column and the other two columns. The least intense washing was done for the 110 cm column with 1 column volume, while the other two columns were subjected to relatively more intense washing with 2-3.9 column volumes. In order to save time on the anyhow quite long methods for the 110 cm column the washing volume was kept lowest. When investigating the different replicates of the 110 cm column, no noticeable differences between first, second and third injection for the 110 cm column could be observed as indicated by very similar error bars as the other two columns. We therefore conclude that the divergent washing between 110 cm column and the other two columns does not substantially impact the presented results. We also want to mention here that the maximum flow rate for the 110 cm column prototype, that was available for us at the time of testing, was only 400nL/min in contrast to 750nL/min for the recent commercial version. We have also now extended Supplemental Table 1 to include all relevant information.

For equilibration, Thermo recommends 1.5 column volumes, since it is mentioned that the superficially porous C18 material on the pillars of μ PACs equilibrates faster than conventional fully porous packed bed columns. We used 2 column volumes for equilibration for all columns corresponding to 9 μ L and 3 μ L for the 110 cm and the two other columns, respectively, regardless of the applied flow. This information has now been added to the manuscript in the results section. Based on the highly reproducible results we obtained for all replicates, we conclude that these 2 column volumes are indeed sufficient for column equilibration.

Sample loading in direct injection mode is a very time-consuming step in comparison to the trap-and-elute setup as the flow rate is >10x lower typically. When the loading volume is set to automatic, which was done in our case, an additional volume of 5 μ L of loading solvent is loaded resulting in a total

volume of around 6 μL . At a flow rate of 400 nL/min, which was the maximum flow rate for the 110 cm column we used, this corresponds to around 15 min of loading time.

The long practical loading times of around 17 min that were typically achieved in combination with the ~ 30 min for equilibration (300 nL/min flow, 9 μL equilibration volume) are the reasons for the 47 min overhead time of the 110 cm prototype column. The section on the overhead times has been rewritten for better understanding and inconsistencies for the run-to-run times of 45 & 60 min gradients for the 5.5 cm column corrected in Figure 4 and Table 2 (former Table 1).

10. On lines 488 & 491 it should be clarified that this is a human:yeast:ecoli proteome mix.

We thank the reviewer for pointing out this potentially unclear statement and have included the information about the triple proteome mix now in the conclusions again.

11. A few more details should be provided in the conclusion/discussion on why the uPAC column outperform the packed columns at the same gradient length? (I guess peak shape/widths?)

Yes, we fully agree with the reviewers' guess. The resulting peak widths are narrower leading to an improved separation power. This is already mentioned in our introduction as we investigated the column properties in an earlier publication. We added an additional statement to the discussion section of the revised manuscript.

12. Could the authors comment on whether the **additional IDs** from WWA impact on better quantification of proteins compared to standard DDA, ideally both in terms of accuracy and precision? Also, how many Chimerys-derived PSMs are actually linked to MS1 features?

We thank the reviewer for this valuable comment. As a measure of precision, coefficients of variation for DDA and WWA are given for low input and single cell samples already in Figure 6. For these examples, WWA achieved similar or slightly better CVs as compared to classical DDA, which indicates that the additional IDs from WWA do not negatively impact quantification. This might be explained by the fact that for many previously identified proteins more peptides get identified with WWA leading to more reproducible quantification for these proteins, while lower abundant proteins uniquely identified by WWA are highly likely to show higher CVs.

The authors appreciate the reviewer's request to include data about the quantitative accuracy of WWA in combination with the CHIMERYs search engine as this is indeed a critical characteristic of any analytical workflow. We have now included an additional panel in Figure 6 as well as an additional supplemental file (Supplemental File 2) showing data of double proteome mixes with varying concentrations of HeLa and yeast digests. The pipeline using WWA indeed allowed the quantification of fold changes between triplicates of samples with less than 10% deviation from the expected fold changes. In fact, WWA showed slightly better quantitative accuracy overall and in particular for fold changes of yeast proteins.

CHIMERYs indeed allows the identification of peptides from MS2 only for which no feature could be detected at the MS1 level. The rationale behind this is that low abundant peptides that cannot be

detected in the MS1 scan due to the limited dynamic range in the C trap and the orbitrap might still be co-isolated and detectable in the MS2 spectrum due to the limited m/z range isolated. We have assessed several files manually and can indeed find a substantial fraction of PSMs for which no MS1 signal intensity is given in the PSM table after searching with CHIMERYS indicating that no MS1 feature could be assigned. We observe 24% of PSMs without assigned MS1 features for a 12.5 ng K562 digest with a 30 min gradient and an isolation width of 1. As may be expected this fraction substantially increases with increasing isolation width and complexity. Also, single cell samples with very little input amount display a considerably higher fraction of PSMs with missing MS1 features. Nevertheless, it can be inferred from Figure 5C that the false discovery rate for CHIMERYS is well controlled for WWA (and DDA) and only a negligible fraction of the additionally identified proteins might present be false positives.

Reviewer #2 (Remarks to the Author):

Thank you for the opportunity to read and review "Micropillar array chromatography, wide window acquisition and AI2 based data analysis enable deep coverage in a broad range of proteomic applications from complex mixtures to limited amount samples".

As the title suggests, there is a lot of data in this manuscript. While that is indicative of the amount of work that is summarized here, it also makes the work, as described, challenging to follow. Highlights for the study include this being one of the first few descriptions of the Chymerys search engine, In addition, there has been relatively little work reported on the openly available IMP-APQuant proteomics quantification algorithm. I find it impressive that the authors are comfortable not only comparing the commercial Chymerys engine to their own well-established search engine MS Amanda, despite the fact that Chymerys appears to outperform their own tools. Given the performance values here it seems that the authors are suggesting the use of the commercial Chymerys engine for spectral library assignment and the use of the freely available APQuant algorithm is an ideal combination. It would be nice to see a note in the text on whether this combination is possible to obtain without purchasing the complete commercial solution from Thermo Fisher. If so, this could be a very powerful option for many proteomics labs at a substantially reduced cost.

The authors thank the reviewer for sharing his/her concerns and suggestions about our manuscript with us. We are grateful to the reviewer for the time and effort spent to help us develop our manuscript further and make it more accessible to the reader. We hope that our answers below and the additions and modifications made to the main manuscript and the supplemental information will justify publication of this manuscript in Nature Communications in the eyes of this reviewer.

Major comments:

The work would benefit from a main text table or schematic that summarizes or provides structure to the work that is described in the text. The very large Excel file (which I believe is Supplemental Data 1 - but the identifiers provided by SpringerLink make difficult for this reviewer to follow) which summarizes 40 cell, single cell and 250 pg peptide load data itself is a very interesting story of optimization for low input reproducible proteomics. The immunoprecipitations provide evidence that this commercial cloud based search engine adds additional biological insight. Again, very interesting, but in this current state this reads more like 3 stories rather than a single cohesive manuscript.

We thank the reviewer for this valuable comment and have now included an additional table (Table 1) in the first section of the results listing the different experiments that were carried out for the manuscript. We also included an additional paragraph highlighting the outcomes and the utilized sample types per experiment in the main text of the results section "Combination of μ PAC with WWA and AI-driven data analysis results in unprecedented proteomic coverage" We hope that this provides additional overview to more easily comprehend the flow of experiments.

We have also included a short paragraph about the applicability of the workflow and that CHIMERYs is unfortunately to the best of the authors' knowledge only available upon purchase.

The data deposited on ProteomeXchange seems to be complete and fully support the findings as described. Each piece of the manuscript is well written.

With something to help clarify and bring structure to the work described here, I'd consider this a work suitable for publication in this journal.

We thank the reviewer for this positive feedback! Of note, new data was included in the resubmission, which is why additional files have been submitted to PRIDE which is detailed in the data availability section.

Minor comments

Line 688 methods: Please clarify the gradient length vs total acquisition time. I think I see what you mean, but I also think it would be helpful for other readers if this was better explained

We thank the reviewer for the valuable comment and have clarified this in the corresponding material and methods section. We have also corrected the gradient time to 2h as this had previously been incorrectly stated.

Figure 7: I don't think there is enough value in Figure 7 for inclusion as a main text figure. I'd suggest moving it to supplemental or inclusion as statements in the text alone.

As also requested by another reviewer, Figure 7 has now been moved to the Supplemental Information as Supplemental Figure 3 and a short paragraph about these data included in the "The AI-driven search engine CHIMERYS substantially improves analysis depth at well controlled FDR using sample input dependent ideal isolation widths" section.

Figure 8: I suspect this figure is too busy to be displayed appropriately in a publication, particularly with text within the Venn diagrams in Fig 8C.

We thank the reviewer for making us aware about this suboptimal figure layout of former Figure 8 that is now Figure 7. We have abbreviated and enlarged the text within the Venn diagrams now and believe that it can be more easily perceived by the reader now.

Supplemental Data 2 (summary of immunoaffinity results) tabs containing Proteome Discoverer output tables could be simplified to remove data not useful to readers. Whether or not the protein was "checked" prior to exporting should be deleted.

We have removed the "checked" column and several other columns in the Supplemental File/Data 3 (former Supplemental Data 2) to simplify the tables while trying to retain all information vital to the readers.

Reviewer #3 (Remarks to the Author):

Summary: The authors present the combination of micropillar array chromatography with wide window analysis and a machine learning based search algorithm to demonstrate deep coverage of low level samples. The authors first comparing packed bed and micropillar columns before evaluating different formats of micropillar columns. The authors then explore various gradient lengths to demonstrate the potential impact of a shorter micropillar column. After choosing the optimal column – the authors compare a machine learning algorithm (CHIMERYS) against a more traditional search engine (MS Amanda 2.0) to demonstrate that CHIMERYS returns more identifications, particularly when using a wide window acquisition technique. The authors then analyze various low level samples and AP-MS experiments to demonstrate the boost in sensitivity of their approach.

The manuscript presents data for a number of different optimizations within their workflow to present an approach that appears to work well for low-level samples. This is a timely topic within the proteomics field as there are a number of new columns, data acquisition techniques, and data analysis approaches that are currently being introduced. One major point I would raise would be the reporting of identified proteins, rather than quantified proteins. For the application of this approach it is important to understand how many proteins can be accurately quantified – rather than just identified. Furthermore, I found figures 3 and 5 to be the most impactful as they provide a well-controlled comparison of the various columns, gradient lengths, and data acquisition strategies. Other figures (e.g., 2, 4, 6) provide intriguing data, but could be expanded upon based on the questions below. Overall, this manuscript would be of interest to the field and the readers of Nature Communications and in my opinion would be suitable for publication following revisions that address questions below.

We thank the reviewer for the support and constructive feedback. The authors highly appreciated the time and effort that this reviewer has invested to help us improve our manuscript. We hope that the answers below and the additions and modifications made to the main manuscript and the supplemental information will justify publication of this manuscript in Nature Communications in the eyes of this reviewer.

Major Points:

1. Protein identification vs. quantification: The figures within the manuscript report the number of identified proteins. However, evaluation of this (and all methods) should comment on the number of proteins that can be accurately quantified as well. In order to utilize this approach to answer meaningful biological questions it will be important to demonstrate that the WWA and CHIMERYS searching algorithm returns additional quantified proteins in addition to identifying more proteins.

We very much agree with the reviewer that this is a very important point and additional protein identifications might not necessarily translate to a deeper understanding of the underlying biology, if these additional proteins cannot be quantified. We very much agree with the reviewer in that we mostly show increased protein and peptide identifications. Next to the already presented number of quantified proteins in Figure 6, we have now changed Figure 7A to display quantified proteins instead of protein identifications showcasing an even bigger relative improvement as compared to the protein ID level. Also, we have added the number of quantified proteins to Figures 7E and 7F again indicating a substantially improved number of identified proteins.

2. Figure 2. Comparison to non-micropillar column: For the comparison of packed bed and micropillar

columns, the variety of packed bed columns could be improved. Specifically, many researchers within the low-level and single cell proteomics field are using IonOpticks columns for their analysis. IonOpticks columns are known to have very high performance, so it would be good to demonstrate the comparison to these columns within this figure.

We appreciate the reviewer's comment and have now also included data for the 15cm IonOpticks Aurora Generation 3 column in Figure 2, which indicate indeed superior performance to the other two previously tested packed bed columns with around 2,000 protein IDs and 10,300 peptide IDs for 12.5ng K562 QC mix. However, the performance of the 5.5cm HT μ PAC Neo column was still found to be superior with roughly 2,200 protein and 12,000 peptide IDs.

3. Figure 5. Optimal window size increases for lower level samples: The authors demonstrate that with decreasing amount of peptide analyzed the optimal WWA window increases. Do the authors believe that this is due gas phase enrichment of precursors that were not detected and triggered on in lower level samples? How do the number of MS/MS events compare between these analyses – are more MS/MS being taken with higher peptides loads? Since WWA depends on the identification of a peak in the MS1 it would stand to reason that smaller windows may not perform well in lower level samples because there are not enough peaks to trigger WWA on a wide region of the precursor m/z range.

We thank the reviewer for raising this important question. We believe that the reviewer's hypothesis is true but only one of the reasons for increased window size with lowered input. With increased window size more precursor ions will be co-isolated and fragmented leading to an increased spectrum complexity. Our data suggests that for higher inputs window sizes of m/z 4 correspond to a sweet spot between being faster and being not too complex for the data analysis algorithm to make a confident identification. With lowered input the lower abundant precursor ions of the very same window would disappear in the noise and would not contribute to an ID but also not to spectral complexity. Therefore here an increased window size is more beneficial to be again able to identify several peptides from one spectrum without having trouble with the resulting spectrum complexity.

Yes, the number of recorded MS/MS correlates to the peptide load and ranges from ~5,500 MS/MS for 250 pg samples to 9,400 for 10 ng samples (30 min active gradient), and from 80,000 for 200 ng to 81,000 for 400 ng (60 min active gradient).

4. Figure 6. WWA improves CVs: Similarly, the authors state that WWA improves CVs though lower injections times. Can the authors demonstrate the number of points across the curve for quantified peptides to demonstrate that WWA changes the number of data points collected – potentially leading to an improvement in quantitation? Is it possible that the WWA doesn't increase the number of the points across the curve, but rather allows peptides to be samples closer to their apex through repeated sampling of particular m/z ranges?

We want to thank the reviewer for his valuable question. During the revision of the manuscript, an inconsistency about the calculations of the CVs for single and 40 cells without MBR was corrected resulting in fewer significant differences between low input sample CVs with DDA and WWA. Significances for analyses with MBR remain identical. Therefore, WWA only provides consistently improved CVs over DDA when MBR is applied. We have investigated the points per peak on the PSMs

level for MBR analyses of our low input and single cell data and found no differences between DDA and WWA, which we now also demonstrate in Supplemental Figure 5. Therefore, the reviewer's suggestion that WWA does not increase the number of datapoints over the peak holds true. While different sampling closer to the apex of the peak seems possible, the authors believe that additional peptides identified and quantified when applying WWA could stabilize protein intensities which takes effect only upon cross-run normalization and matching as observed in the MBR analysis.

5. Figure 7. Broader dynamic range with WWA: This figure does not provide sufficient evidence that the dynamic range is significantly different with WWA. I would either remove this figure or collect additional data to demonstrate that the dynamic range is significantly different (with replicates, etc.).

We agree with the reviewer that we do not see an effect that changes the dynamic range over orders of magnitude, but indeed the upper limit of detected peptides was shifted by up to 100%. At the lower end of low abundant peptides, the observed effect is small but still well pronounced. That said, we believe that this dataset might be of interest, even though the effect is not extremely high, also in order to provide a comprehensive picture to our readers. To oblige with the reviewer's request, we decided to move the figure to the supplementary material. We also shortened and integrated the accompanying text into the chapter "The AI-driven search engine CHIMERYS substantially improves analysis depth at well controlled FDR using sample input dependent ideal isolation widths" within the revised manuscript.

Minor Points:

1. Figure 4. Proteins per minute: This may be a personal preference – but protein IDs per minute of total runtime is not too useful of a metric for those looking to understand how many samples could be run in a day while maintaining a certain proteome depth. It would be interesting to see these data plotted as the number of samples that could be analyzed per day with a depth of X proteins (e.g., 4,000 or 6,000 proteins). This would help to demonstrate the differences in the throughput without readers needing to do the math in their head.

We want to thank the reviewer for this comment and agree that this information could be highly interesting for the readers of the manuscript and have therefore included the requested information in the main text of the results section.

REVIEWERS' COMMENTS

Reviewer #1 (Remarks to the Author):

I appreciate the authors' lengthy response to our comments and questions, and would like to thank them for their efforts to explain the uncertainties we had with the initial version of the manuscript. All our concerns have been addressed and we would recommend publication of the manuscript in its current form.

Reviewer #3 (Remarks to the Author):

The authors have substantially improved the manuscript. In particular there was an increased emphasis on quantitative accuracy, rather than just identification of proteins. In addition to this concern, my other concerns were met or answered to my satisfaction. I support publication of this revised manuscript.